



# Seasonal timeline for snow-covered sea ice processes in Nunavik's Deception Bay from TerraSAR-X and time-lapse photography

Sophie Dufour-Beauséjour[1,2], Anna Wendleder[3], Yves Gauthier[1,2], Monique Bernier[1,2], Jimmy Poulin[1,2],
Véronique Gilbert[4], Juupi Tuniq[5], Amélie Rouleau[6], Achim Roth[3]

[1]Centre Eau Terre Environnement, Institut national de la recherche scientifique (INRS), Quebec, G1K 9A9, Canada
[2]Centre d'études nordiques (CEN), Université Laval, Quebec, G1V 0A6, Canada
[3]German Aerospace Center (DLR), Oberpfaffenhofen, 82234 Weßling, Germany
[4]Kativik Regional Government, Kuujjuaq, J0M 1C0, Canada
[5]Salluit, J0M 1S0, Canada
[6]Raglan Mine (a Glencore Company), Laval, H7S 1Z5, Canada

*Correspondence to*: Sophie Dufour-Beauséjour (sophie.dufour-beausejour@ete.inrs.ca)

**Abstract.** Inuit have reported greater inter-annual variability in seasonal sea ice conditions. For Deception Bay (Nunavik), an area prized for seal and caribou hunting, an increase in solid precipitation and a shorter snow cover period is expected in the near future. In this context, and considering ice-breaking transport in the fjord by mining companies, we monitored sea ice in the area for three seasons of ice between 2015 and 2018. This article presents a case study for the combined use of TerraSAR-X and time-lapse photography time-series in order to monitor snow-covered sea ice seasonal processes. The X-band median backscattering is shown to reproduce the seasonal evolution expected from C-band data. Two different freeze-up and breakup processes are characterized. New X-band backscattering values from newly formed ice types are reported. The monitoring approach presented in this article has the potential to be applied in other remote locations, and processes outlined here may inform our understanding of other fjords or bays where ice-breakers transit.

## 1 Introduction

### 1.1 Context

Salluimiut (people of Salluit, Nunavik, in Canada) have reported changes in their environment which affect activities on the land in Deception Bay (Tuniq et al., 2017). This area is prized by local Inuit for fishing, as well as seal and caribou hunting (Petit et al., 2011). People from neighboring community Kangiqsujuaq have reported warmer and longer fall seasons, later freeze-up, and thinner ice (Nickels et al., 2005), as well as less snow and earlier sea ice breakup in spring (Cuerrier et al., 2015). The evolution of seasonal sea ice conditions in Deception Bay is expected to continue, with climate projections for the region showing shorter snow cover periods and an increase in solid precipitation (Mailhot and Chaumont, 2017). Further, two nickel mines have marine infrastructure in Deception Bay. Their ice-breakers transit in the bay from June 1st to mid-March, avoiding the seal reproduction period (GENIVAR, 2012). Monitoring seasonal snow-covered sea ice processes in the area is relevant in light of local community members' reliance on the fjord's rich ecosystem for subsistence, as well as for shipping-related operations by the mines. Lessons learned from this work further have the potential to be applied in similar contexts elsewhere in Inuit Nunangat (Inuit regions of Canada, including Nunavik), and by other researchers studying the cryosphere, particularly those using X-band radar remote sensing.



### 1.2 Monitoring snow-covered sea ice

General first-year sea ice processes include formation through freeze-up, transformation of the snow and ice covers, and eventual breakup. These processes may unfold differently from year to year due to the weather, over a period of time which may vary from a single day to weeks. They may be driven by environmental factors such as air temperature, wind, currents, and precipitation, to name several. The sequence of events further depends on geomorphological features like shallows or deep water pockets, an island or a river. In order to establish a seasonal timeline of snow-covered sea ice processes, it is therefore necessary to rely both on spatial coverage of the bay as well as frequent observations. These requirements are met by the combined use of radar remote sensing and time-lapse photography.

Time-lapse photography is well-suited to high-repeat-time monitoring applications in Inuit Nunangat, as well as polar regions in general: the systems can be installed in remote locations and record data, as often as hourly, for prolonged periods of time. Such time-series have been used to track daily-to-seasonal variations in the extent of the sea ice and ice melange in front of a retreating glacier (Cassotto et al., 2015), to document glacier mass loss (Chauché et al., 2014), and to observe sea ice concentration in the Beaufort Sea (Wobus et al,. 2011). Time-lapse photography has also been used to document snow accumulation and accretion processes on mountain slopes (Vogel et al., 2012), snow cover extent in the tundra (Kepski et al., 2017) and in forests (Arslan et al., 2017), as well as snow melt (Farinotti et al., 2010; Ide and Oguma, 2013; Peltoniemi et al., 2018; Revuelto et al., 2016). Meteorological information may be derived from the photographs, for instance precipitation type or wind conditions (Christiansen, 2001; Liu et al., 2015; Smith Jr et al., 2003). Finally, time-lapse photography time-series have been used to validate interpretations of active radar remote sensing images in the context of iceberg plumes and coincident sea ice conditions (Herdes et al., 2012).

Synthetic aperture radar (SAR) sensors are uniquely qualified for winter applications in Inuit Nunangat since they can acquire images in the dark and through clouds. Modern remote sensing options combine wide coverage and high spatial resolution with a revisit period as short as 11 days, in the case of TerraSAR-X (X-band, 9.65 GHz). Scatterometers offer the advantage of continuous measurements, but ship-based deployments are prone to ice and ship drift, while laboratory environments lack naturally occurring phenomena such as heat fluxes and currents (Isleifson et al. 2014). Recent studies have taken advantage of TerraSAR-X's frequent revisits to successfully document spatially extensive processes such as seasonal snow cover extent and snowmelt (Sobiech et al., 2012; Stettner et al., 2018), as well as glacier calving front monitoring (Zhang et al. 2019). In the context of first-year sea ice, a substantial ERS-1 and RADARSAT-1 (C-band, 5.405 GHz) time-series spanning 8 years was aggregated to study the springtime backscattering signature of snowmelt processes on landfast ice (Yackel et al., 2007).

SAR X-band has been shown to be a useful complement to the conventional C-band when it comes to first year sea ice; it was used to identify types of new ice (Johansson et al., 2017), particularly thin ice like nilas and grey ice (Matsuoka et al., 2001). The X-band is reputed to be more sensitive to the snow cover as well as to freeze/thaw processes than the C-band (Eriksson et al., 2010). Yet, an inventory of the literature presented by Johansson et al. (2017) shows a gap in observations—only one backscattering signature is reported for newly formed ice in the X-band (Nakamura et al., 2005). Several recent publications have increased our understanding of the X-band backscattering signature of first-year sea ice: values were reported for new ice and nilas (Johansson et al., 2017; 2018) and for white ice (Fors et al., 2016) during the winter, as well as for first-year sea ice during the spring (Nandan et al. 2016; Paul et al., 2015). Despite these advances, the majority of the literature on radar remote sensing of first-year sea ice is in the C-band, and understanding of the X-band signal often has to be deduced from analogy with the latter rather than from direct observations.





### 1.3 Objectives

This article investigates the X-band backscattering signature of snow-covered first-year sea ice while contextualizing the reported data in observed seasonal processes, which serves several purposes. First, the monitoring approach shared in this article has the

80 potential to be applied in other remote locations of local or broader scientific interest. Second, processes outlined here may inform our understanding of other fjords or bays where ice-breakers transit. This study stands out due to the length and continuity of the time-series reported, its use of an X-band sensor, and its relevance to local actors. Specifically, the objectives of this article are to 1) investigate the hypothesis that the seasonal evolution of X-band backscattering from snow-covered first-year sea ice will follow that of the C-band; 2) document freeze-up and breakup processes for a fjord in which there is ice-breaking transport; 3) contribute

a new dataset on the X-band backscattering signature of newly formed sea ice.

### 2. SAR backscattering over snow-covered first-year sea ice

Due to the relative lack in X-band-specific work on scattering mechanisms in snow-covered sea ice, the following brief literature review focuses on C-band backscattering. Both bands are however close in terms of frequency (or wavelength), with TerraSAR-X's X-band centred at 9.6 GHz (3.12 cm) and RADARSAT-2's C-band at 5.405 GHz (5.55 cm). General observations on scattering

in the C-band are followed by expected differences in behavior between the X- and C-band. We focus on freeze-up, winter, and spring. We also quickly remind the readers of the World Meteorological Organization (WMO) sea ice nomenclature relevant to our study, particularly first year sea ice types (WMO, 2014).

Several scattering mechanisms are significant in snow-covered sea ice. Firstly, surface scattering may occur on seawater or brine,

at the air-ice interface, at the air-snow interface if the snow is wet, at the interface between dry and wet snow, and finally at the snow-ice interface. Secondly, volume scattering may occur within brine-wetted snow as well as within the ice. The important scatterers are therefore brine and water inclusions, both within snow and ice. Figure 1 shows sketches of snow-covered sea ice from freeze-up to winter and spring, including the presence of these scatterers. The information depicted is based on the literature. The following subsections start with a description of physical processes related to these scatterers, which is followed with a

presentation of the associated backscattering mechanisms.

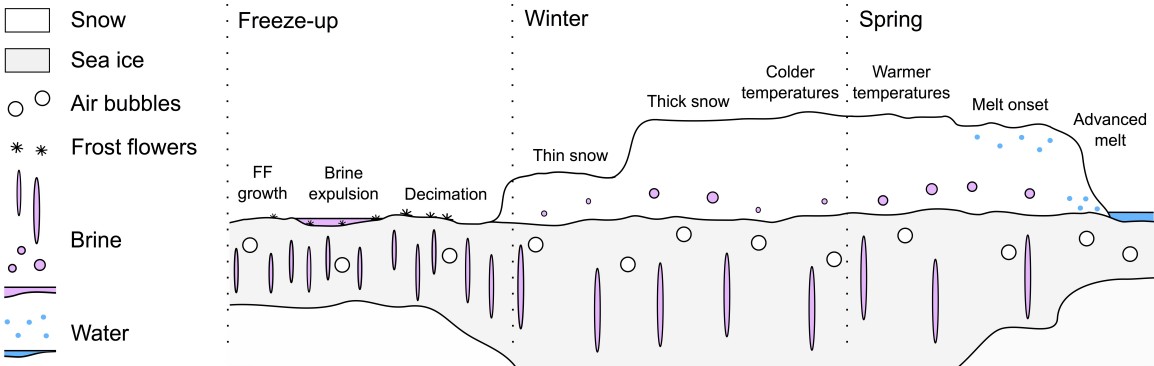

**Figure 1:** Sketch of scatterers in snow-covered sea ice from freeze-up through winter and spring. Shown: frost flowers (FF, stars), air bubbles (white circles), brine inclusions in the ice (pink elongated ovals), liquid brine at the surface of the ice (pink), brine

inclusions in the snow (pink circles), water inclusions in the snow (blue circles), and melt ponds (blue). Each season is described in Sect. 2.1 to 2.3.



### 2.1 Freeze-up

The youngest form of sea ice is designated in WMO (2014) nomenclature as "new ice", a term which includes ice types where ice crystals are weakly frozen together, like shuga, frazil and grease ice. Other newly formed ice types include nilas, matt flexible ice thinner than 10 cm, and ice rind, shiny brittle ice thinner than 5 cm (ibid.). Finally, circular pieces of ice up to 3 m in diameter may form from grease ice, or from broken nilas or ice rind. This ice type, called pancake ice is typically less than 10 cm thick (ibid.). Ice growing older and thicker than 10 cm is designated as young ice, a term which includes grey ice (10 to 15 cm thick) and grey-white ice (15 to 30 cm) (ibid.).

As ice forms from seawater, brine is entrapped within the frozen water crystals (Cox and Weeks, 1988)—first-year sea ice is a very lossy material for which surface scattering dominates (Onstott 1992). New ice formed in calm conditions may give an almost mirror-like return due to its smoothness, while rafting, snowfall, or frost flower formation will all increase the surface roughness and thus the backscattering (ibid.). Backscattering in the C-band has been shown to increase in the presence of frost flowers, peaking at the height of their growth (Isleifson et al. 2014). A small fraction of brine is expulsed at the surface of the ice in the hours following freeze-up and may be wicked up by frost flowers, potentially decreasing backscattering (ibid.). In the absence of frost flowers, a post-freeze-up backscattering peak may still be observed, attributed to an increase in brine drainage channel size during the early desalination process (Nghiem et al., 1997). By the time the ice is roughly 40 cm thick, most of its brine will have drained out through vertical channels under the effect of gravity or thermal gradients (Cox and Weeks, 1974; Zhang et al., 2013). At this stage of grey-white ice, any frost flowers will have been infiltrated with snow or decimated, yielding a decrease in backscattering (Onstott, 1992). Snow-covered frost flowers may still contribute to surface scattering (Isleifson et al. 2010). The higher-frequency X-band is more sensitive to surface roughness than the C-band (Eriksson et al., 2010), and allows better discrimination between different thin ice types (Johansson et al., 2017).

### 2.2 Winter

Ice which is between 30 and 70 cm is described as "white ice" (ibid.), and first-year ice thicker than 70 cm is simply called "first-year ice" (ibid.). Gill et al. (2015) presented a synthetic review of scattering mechanisms in snow-covered sea ice; the following description borrows from their structure. Following freeze-up, snow starts to accumulate on the ice cover, absorbing brine from the ice through capillary action (Barber and Nghiem, 1999), which will remain in the bottom snow layer as brine inclusions. No surface scattering is expected at the air-dry snow interface (Kim et al.. 1984), and volume scattering from dry snow is small (Kim et al., 1984). However, the presence of brine-wetted snow between dry snow and ice has a significant impact on backscattering.

Firstly, the dielectric mismatch between the dry and brine-wetted snow layers may cause surface scattering (Nandan et al., 2016). Secondly, volume scattering may occur on brine inclusions within the snow (Gill et al., 2015). As the ice cover thickens, the insulation it provides from the water allows temperatures at the snow and ice surface to drop (Yackel et al., 2007). The beginning of a stable backscattering signal marks the onset of the winter regime (ibid.), which may feature a decreasing trend as brine inclusions shrink under colder temperatures (ibid., Nghiem et al., 1997). For a given temperature, thick snow covers will induce higher temperatures at the snow-ice interface, which will increase the volume of brine inclusions therein (Barber and Nghiem, 1999), and thus the backscattering (Gill et al., 2015). Surface scattering from wet snow may completely mask the underlying snow and/or ice layers (ibid.), and lead to a backscattering signal dominated by the wet layer and layers above. If penetration in the sea ice occurs, the total backscattering may include a similarly temperature-dependant contribution from volume scattering on brine inclusions in the ice (Barber and Nghiem, 1999).





Penetration depth in the brine-wetted snow is lower for the X-band than for the C-band (Nandan et al., 2016), due to the former's shorter wavelength. For incidence angles between 35° and 48°, our range of interest for this study, X-band backscattering should be dominated by surface scattering on the brine-wetted snow layer and decrease with incidence angle (Nandan et al., 2016). For

the C-band in the same range, volume scattering in the brine-wetted snow is expected to dominate the total backscattering (ibid.).

### 2.3 Spring

As described by Yackel et al. (2007), the early melt period which precedes melt onset is known to present diurnal variations in snow temperature, water content, and brine volume which may yield different backscattering values for ascending and descending acquisitions, for instance lower backscattering in the afternoon when compared to nighttime acquisitions (Barber and LeDrew,

1994). Melt onset is characterized by an increase in snow and ice brine inclusion size brought on by warmer temperatures, which yields an increase in backscattering (Barber and Nghiem, 1999). Above-zero temperatures and solar radiation following melt onset can also lead to humidity in the top of the snowpack (Gogineni et al., 1992; Kim et al., 1984), yielding surface scattering from the wet snow (Gill et al., 2015; Yackel et al., 2007). The X-band is expected to be more sensitive to melt onset than the lower-frequency C-band (Eriksson et al., 2010). The presence of a wet snow layer at the top of the snowpack will act in a similar way as the brine-

wetted snow layer described earlier, i.e. potentially masking the underlying snow and ice (Gill et al., 2015).

Water can be held in the snowpack (pendular regime) up to a 7% water content threshold (Scharien et al., 2012). Past this limit, meltwater from the snow cover will drain down to the ice surface (ibid.), flushing out brine (Barber et al., 1995), potentially refreezing if temperature drops below 0°C (Gogineni et al., 1992). The transition from water being held in the snowpack to this

funicular regime where water drains downwards, called "pond onset" (Yackel et al. 2007), is associated with a decrease in backscattering (Barber et al. 1995) which may or may not be observed depending on snow thickness before melt (Yackel et al., 2007). In this period known as early advanced melt, ponds form where the now-isothermal snow is thin (Scharien et al., 2012). In the peak melt phase, the snow has completely melted and pond coverage reaches a maximum, before melt progresses to its late stage where the ice becomes permeable (ibid.). Melt water then drains rapidly from the ice, before complete decay or breakup

(ibid.). The backscattering from sea ice covered by melt ponds will depend on the wind conditions (ibid.).

### 3. Study area

Deception Bay (62° 09' N, 74° 40' W) is located on the northern edge of Nunavik, the Inuit Nunangat territory overlapping the Canadian province of Quebec north of the 55th parallel. This fjord of the Ungava Plateau is roughly 20 km long, and nested in hills peaking at 580 m in altitude (GENIVAR, 2012). Water depth in the bay (Fig. 2) reaches 80 m in the deepest section located between

the marine infrastructure and Moosehead Island. Deception Bay is accessible from Hudson Strait by boat during the ice-free season, or by icebreaker. It is also accessible in winter and spring by snowmobile from overland trails.

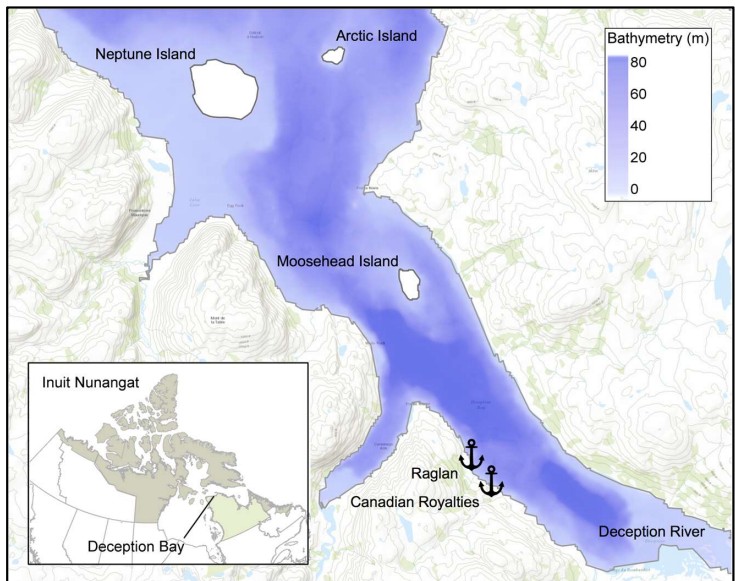

**Figure 2:** Elevation and bathymetry map of Deception Bay. Inset: Inuit Nunangat, with Nunavik in green. Marine infrastructure
identified with harbor markers.

The Canadian Ice Service, in its *"Climatic Ice Atlas 1981-2010"*, estimates freeze-up and breakup in the bay to occur around the
first week of December and the first week of July, respectively (Fequet et al., 2011). Landfast sea ice typically extends to the mouth
of the bay, where it is stabilized by Neptune Island. Snow thicknesses measured in Deception Bay at the end of April 2017 ranged
from 0 to 31 cm while observed ice thicknesses were between 1.11 and 1.45 m (Gauthier et al. 2018). Dominant winds come from
the southwest (GENIVAR, 2012). Although Environment Canada operated a weather station in Deception Bay from 1963 to 1973,
there is currently no weather station in the bay. The closest one is located at Salluit airport, a similar coastal environment 50 km
west of Deception Bay. Mean and high tides in the bay have respective ranges of 3.9 and 5.7 m. Deception River is the largest
river flowing into the bay, and its flow is greatest at the end of spring in June and July because of snow melt. Deception River flow
is almost zero during the winter (ibid.). Water salinity in the bay ranges from 29 to 33 psu (ibid.).

The Northern Villages of Salluit and Kangiqsujuaq and both communities' Land Holding Corporations were contacted and gave
their approval for this project, including associated activities and instrumentation in Deception Bay. The Avataq Cultural Institute
was consulted to ensure the project didn't encroach on archeological sites important to Inuit. Finally, the Nunavik Marine Region
Impact Review Board was contacted to get permission for the deployment of underwater sonars in Deception Bay (sonar data not
presented in this article).

**4. Data description**

**4.1 Time-lapse photography**
A pan-tilt-zoom Panasonic WV-SW598 camera was installed on the south-west shore of Deception Bay (Fig. 3) on 11 September
2015. Operating in time-lapse mode, the camera takes a photograph every 15 minutes during the day (from 6:00 to 18:00 LT),
rotating through four preset views (Fig. 3). The photographs have an effective pixel count of 2.4 megapixels and a 90x zoom is
available when setting the views or taking remote control of the camera. The camera can operate at temperatures between -50°C





and 55°C and is installed at a height of 1.8 m. The structure is composed of a metal mast welded to a 50 cm wide square metal base bolted in the bedrock. The selected site is accessible by foot from Raglan's marine infrastructure, located on a high-point

which offers a good view of the bay, and is in front of a Raglan power and network access point. This allows systematic transmission of the photographs to a database hosted by INRS. There are roughly 1 400 photographs per month, for a total of almost 17 thousand per year, all available to the general public on http://caiman.ete.inrs.ca (Bernier et al., 2017).

Two Reconyx PC800 Hyperfire Professional Semi Covert cameras were similarly installed in Deception Bay as part of the

CAIMAN research project. These cameras were installed in front of Moosehead Island (series A) and on Black Point (series B) and rely on 12 V batteries and solar panels for power, as shown on Fig. 3. They measure the temperature within the camera case and record this information in the photographs' metadata. In this article, only their temperature measurements are used and not the associated photographs, since their field of view is outside the area of interest. The temperature measurement is susceptible to the following sources of error: camera heating from the sun and the absence of data between 19:00 and 6:00. Despite this, we choose

to use this local measurement—corrected for camera heating—instead of the data measured 50 km away, in another bay and further inland (airport) than the cameras (on the shore).

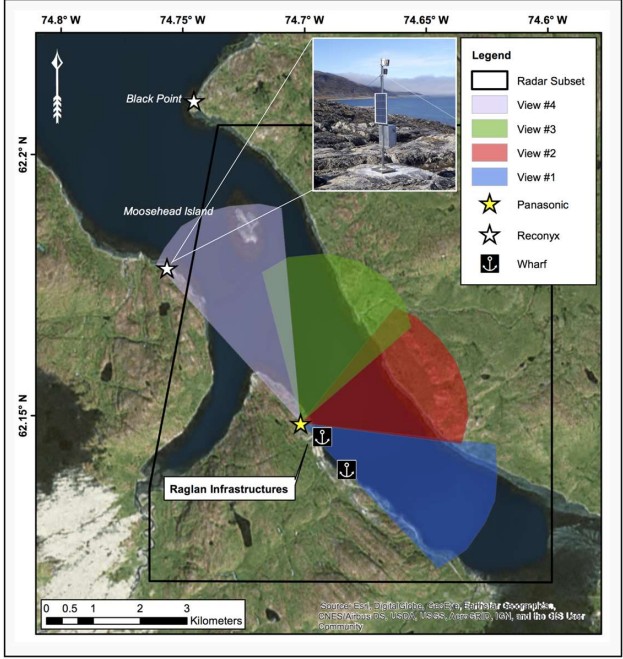

**Figure 3:** Map of time-lapse camera locations and Panasonic fields of view. Inset: Reconyx camera on the south-western shore in
front of Moosehead Island (solar panel and battery also shown).

### 4.2 TerraSAR-X

TerraSAR-X StripMap dual co- and cross-polarization single look complex (SLC) images were acquired over Deception Bay from December 2015 to July 2018, spanning three winter seasons (see Table 1 for acquisition characteristics). This X-band satellite— and its counterpart TanDEM-X—operate at a central frequency of 9.65 GHz (3.11 cm wavelength), with a repeat period of 11

days. Three orbits were used for acquisitions (13, 21, 89)—orbits 21 and 89 are respectively one and five days later than orbit 13.



They cover a range of incidence angles between 38° and 46°, include both ascending and descending passes, and all share a VV polarization. The scene size before subsetting was 15 by 50 km, with a spatial resolution of 0.9 and 2.5 m respectively for range and azimuth directions (Eineder et al., 2008).


**Table 1:** Characteristics of TerraSAR-X acquisitions for the study.

| Orbit | Local acquisition time (UTC -5:00 / -4:00) | Incidence angle | Polarisations | Acquisition period | Total number of images |
|---|---|---|---|---|---|
| 13 | 17:32 (ascending) | 38° | HH/VV | 23 December 2015 to 26 July 2018 | 75 |
| 21 | 6:25 (descending) | 40° | VV/VH | 24 December 2015 to 27 July 2018 | 70 |
| 89 | 17:40 (ascending) | 46° | VV/VH | 28 December 2015 to 31 July 2018 | 76 |

## 5. Methods

### 5.1 Visual inspection of photographs

The photographs were visually inspected as needed to document freeze-up and breakup. Figure 4 shows the four views of the Panasonic camera during freeze-up 2016, when Deception Bay is covered with nilas up to Moosehead Island (indicated with a red arrow in Fig. 4). We define freeze-up as the day when ice on the bay—which may include different types of newly formed ice such as grease ice, nilas, etc.—consolidates into a continuous cover with no lateral movement induced by wind or current, and stays in place for the whole winter. The bay may very well be largely covered by ice on some occasions before formal freeze-up occurs.

Visual inspection focused on sea ice extent over the four views, the persistence of features in the sea ice over time, and their lateral movement or the absence thereof. The breakup process was similarly characterized by identifying the first occurrence of open water somewhere in the fields of view of the camera, and documenting the progression of its extent over time up to a completely ice-free state. The day of breakup is the first day where the bay is ice-free.

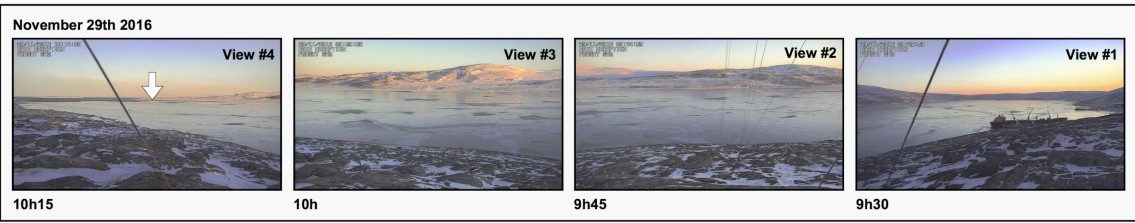

**Figure 4:** Time-lapse photography panorama of Deception Bay during freeze-up 2016. Photographs were taken on 29 November 2016 from the south-western shore at 15-minute intervals looking south, then south-east, east and finally north-east. Moosehead Island is identified with an arrow, for reference.

### 5.2 From temperature to freezing and thawing degree-days

Temperatures time-series were assembled from photograph metadata for two cameras: Moosehead Island (Series A) and Black
Point (Series B). Hourly temperatures were available daily between 7:00 and 18:00 LT. Series A was continuous over the study period except for a gap between 27 January and 18 September 2016, while series B stops on 16 September 2016. The two series



were compared for the overlapping period (11 September 2015 to 27 January 2016). The difference between series A and B was 0.4 °C on average, with a standard deviation of 0.7 °C. This was deemed sufficiently small to combine the two series with no transformation. Series B was used from 11 September 2015 to 16 September 2016, and series A from 18 September to 31 August

2018. Since the Reconyx temperature sensor sometimes erroneously recorded a 0°C measurement in lieu of "not-a-number", all 0 °C values were removed from the datasets. Daily mean temperature was computed from data between 7:00 and 18:00. In December, January, and February, the sunrise occurs after 7:00 and the sunset before 18:00. The daily mean therefore presents a bias towards daytime temperatures which evolves throughout the year as a function of sunrise and sunset time. Since this bias is the same from year to year, it does not affect inter-annual comparisons like the ones presented in this article.


A second and more significant source of bias is found in camera heating by the sun, which increases measured temperatures. As part of another study, Reconyx cameras are also installed in communities where air temperature is measured hourly at the airport by Environment Canada (EC). In the supplementary materials we compare the camera-measured temperature with the EC reference for Quaqtaq, located 300 km south-east of Deception Bay (Fig. S1-S2). The difference between the two temperatures exhibits a

similar seasonal dependence over the years: low from September to January, and significant from February to August. From this the heating-induced bias to the temperature measured by the camera is modelled as a Gaussian function with 4°C amplitude and a sigma of 50 days, and subtracted from the camera data to yield a corrected time-series. The camera heating bias was roughly centered on May 1st for the three years used to develop the simple model. It may however shift from this date depending on the year, as might have been the case for Salluit and Deception Bay in 2015-2016 (ibid.). When a reference is available, we recommend

centering the modelled bias by comparing the camera data with the reference. In this study however, we chose to center the modelled bias on May 1st in order to explore the use of the camera measurements for cases where there is no neighboring reference station. Data before and after the correction is presented in the supplementary materials, along with its comparison to the data from 50-km distant EC station at Salluit airport (ibid.). After correction for the camera heating, the Deception Bay temperature is still 2°C higher on average than the EC data from Salluit airport. This is attributed to differences in altitude relative to sea level and in

distance from the shore between the two measurement locations.

Two sets of indicators are derived from the temperature time-series: the first freezing degree-day and the first thawing degree-day, as well as cumulative freezing or thawing degree-days at the time of freeze-up or breakup. Freezing and thawing degree-days are defined in Eq.(1):

$FDD(T) = 0 \ if \ T > 0, |T| \ if \ T \leq 0$             (1a)
$TDD(T) = T \ if \ T \geq 0, 0 \ if \ T < 0$             (1b)

where FDD is the freezing-degree day, TDD is the thawing degree-day, and T is the daily mean temperature (NSIDC, 2019). The cumulative sums of freezing and thawing degree-days over a period—CFDD and CTDD respectively—are used to characterize how cold or warm that period is (ibid.; Permafrost Subcommittee, 1988). In the absence of temperature data between 19:00 and

6:00, the daily mean computed from the time-series is biased towards daytime temperatures: freezing and thawing degree-days computed using Eq.(1) may therefore also be biased.

### 5.3 TerraSAR-X image processing and analysis

Processing of the TerraSAR-X images was done at DLR, the German Aerospace Center, using the Multi-SAR System. This workflow starts with a conversion from the digital number to radar brightness (beta-naught), followed by multi-looking to produce

square pixels and increase the radiometric quality (number of looks), orthorectification so all the images from all orbits could be



overlaid, and image enhancement to reduce the speckle inherent to SAR images (Schmitt et al., 2015). The Multi-SAR System is described in detail in Bertram et al. (2016). The output images have a geometric resolution of 2.5 m pixels with a radiometric resolution of 1.6 looks. The TerraSAR-X noise floor for the three orbits ranges between -23 and -24.5 dB, and the radiometric accuracy is 0.6 dB (Eineder et al. 2008).


Median backscattering was computed for each image over 32 areas of interest (AOIs) distributed over a portion of the bay common to all orbits and identified on Fig. 3. These AOIs, roughly 120 m by 100 m, contain between 2016 and 2064 pixels each. Their locations were chosen to avoid the shore, man-made structures like docks, as well as broken ice left in the wake of ice-breakers (Fig. 5). This step was performed using Python (Dufour-Beauséjour, 2019).


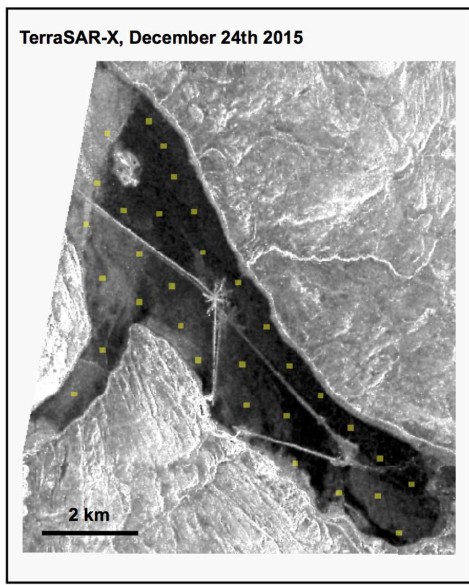

**Figure 5:** TerraSAR-X VV image of Deception Bay on 24 December 2015 in orbit 21 (scaled from -19 to -5 dB) and AOIs used to compute statistics (yellow).

The median backscattering over first-year sea ice features two seasonal peaks separated by two inflexion points: the post-freeze-up peak, the beginning and end of the monotone period, and the spring peak (see Fig. 6), which are respectively associated with frost flower maximum, winter onset, melt onset, and pond onset (see Sect. 2). Speaking in terms of the data time-series, peak location is defined as the location of its maximum, and estimated as sitting between the left and right-hand neighbors of the highest data point. The beginning of the monotone period was estimated as sitting between the first monotone data point and its left-hand

neighbor; the end of the monotone period was similarly estimated from the last monotone data point its right-hand neighbor. Peak and inflexion point locations for all orbits and years are presented in the supplementary materials (Fig. S3-S5). The location of each feature was estimated manually and given as a range (see color-shaded areas in Fig. 6), which was further reduced by combining all available orbits (Fig. S3-S5). Finally, the winter trend was computed from a linear regression fit on the data in the monotone period, as shown in the supplementary materials (Fig. S6).






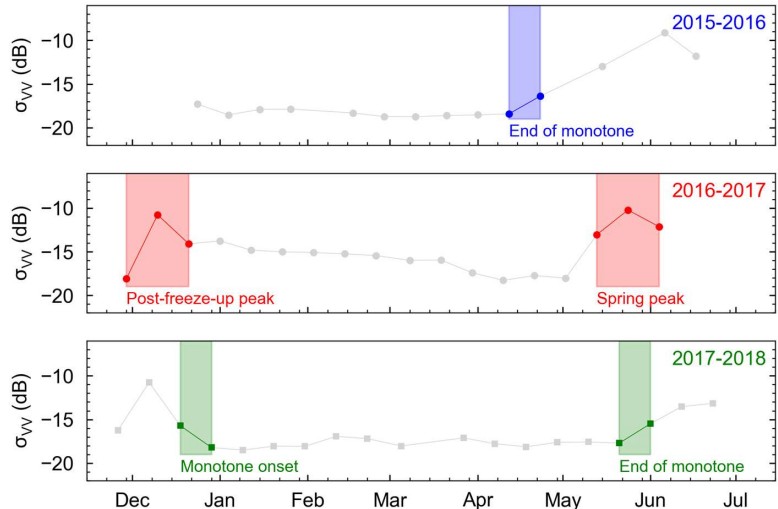

**Figure 6:** Examples of change detection in TerraSAR-X VV median backscattering. Inflexion detection for orbit 21 in 2015-2016 (top), peak detection for orbit 21 in 2016-2017 (middle), and inflexion detection for orbit 13 in 2017-2018 (bottom).

## 6. Results

### 6.1 Freeze-up

Freeze-up was documented using time-lapse photography and TerraSAR-X images. The daily event sequence (Tables S1-S3), as well as freeze-up videos assembled from time-lapse photography (Movies S1-S3), are both available as supplementary materials. In the following description of each year's freeze-up, we refer to zones represented in Fig. 7. No ice-breaker transits occurred during freeze-up for the three years of this study.

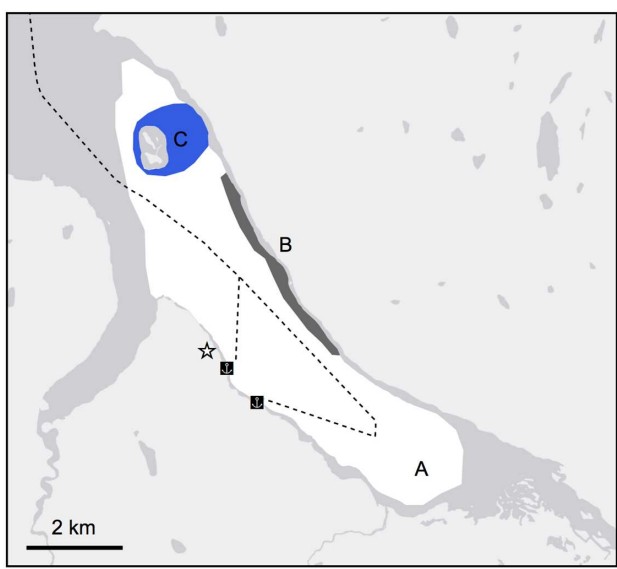

**Figure 7:** Zones relevant for describing the spatial aspects of freeze-up in Deception Bay, and ship routes for the MV *Arctic* and MV *Nunavik* (dashed line). Camera location is indicated with a star.



In 2015, the days before freeze-up featured fog and open water, as well as grease ice and shuga. Freeze-up began on 10 November, where landfast ice first appeared along the north-eastern shore (zone B) and grew thermally, progressively extending to cover the
whole bay (zone A) by 11 November 2015. In 2016, the days before freeze-up featured grease ice and open water, and the accumulation of pancake ice over the shallows near Moosehead Island (zone C). After the formation of nilas and various new ice types on 27 November (Fig. 8a), zone A was covered in mirror-like patches of nilas and ice rind on 28 November. Their lateral movement is illustrated in Fig. 8b. The next morning, the bay was similarly covered with overlapping patches of nilas. No lateral movement of the ice was observed on 29 November, as shown in Fig. 8c. Freeze-up was therefore completed on 29 November
2016. In 2017, the series of events was the same as the year before. Freeze-up was alternatively preceded by days of open water and days where the bay was covered in grease ice or nilas, and pancake ice accumulated in zone C. On 27 November, zone A was covered with mirror-like nilas or ice rind. This ice was rearranged during the night into an ice cover which showed no further substantial lateral movement. Observed features shifted slightly south-east in the night between 29 and 30 November. Even so, we identify freeze-up as having occurred on 28 November 2017.


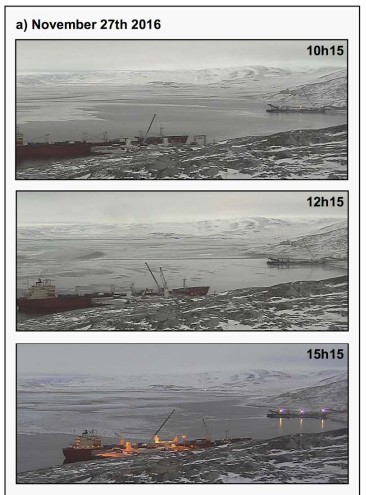
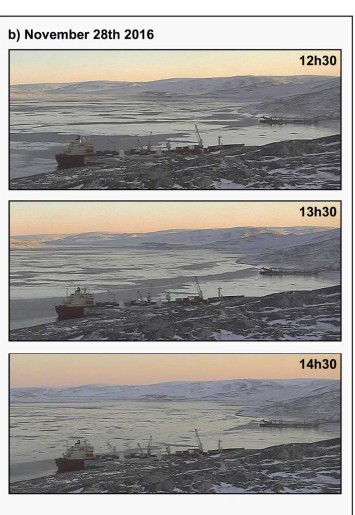
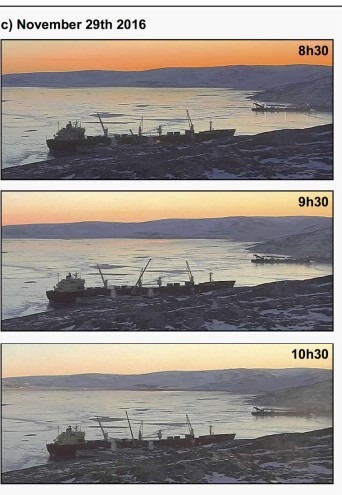

**Figure 8:** Time-lapse photography of Deception Bay during freeze-up 2016. Photographs were taken from the Southeastern shore on a) 27 November 2016, open water and patches of nilas moving with time, b) 28 November 2016, idem, and c) 29 November 2016, a stable nilas ice cover.


TerraSAR-X acquisitions during freeze-up 2016 and 2017 were used to extract the X-band backscattering signature of newly formed ice types, identified from time-lapse photography. Figure 9 shows an example from 26 November 2017, where grease ice was observed as well as a mix of nilas and pancake ice. Median values for newly formed ice types are presented in Fig. 10. Two images featured nilas, coincidently with pancake ice. Five images of grey-white ice were acquired. These results are presented for
different acquisition geometries and incidence angles. The images associated with each box in Fig. 10 are reproduced in the supplementary materials (Fig. S7-S8), along with the color-coded AOIs used for each ice type.



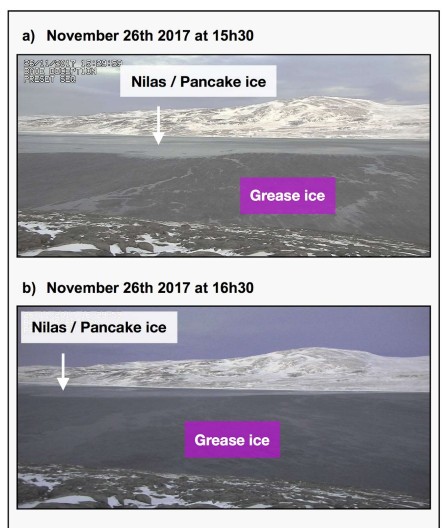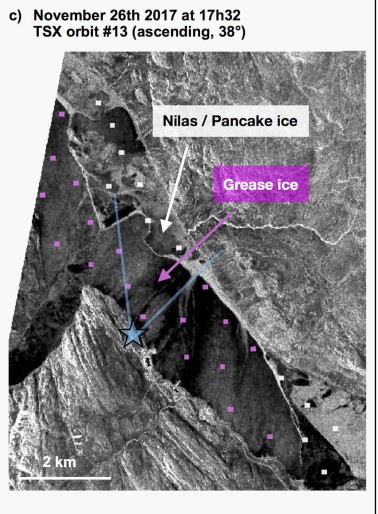

**Figure 9:** Grease ice and a mix of nilas and pancake ice in Deception Bay on 26 November 2017. a) and b) Photographs taken from the shore showing grease ice in the forefront occupying most of the field of view, and the lighter and more uniform

arrangement of nilas and pancake ice in the background. c) TerraSAR-X VV image from orbit 13 (scaled from -19 to -5 dB). AOIs are color-coded according to the ice type at their locations. Solid ice is consolidated against the northern shore and around Moosehead Island (top left of image), composed of nilas and pancake ice (white squares). The rest of the bay was covered in grease ice (pink squares). The location of the camera is identified with a blue star and its field of view limits indicated with blue lines.


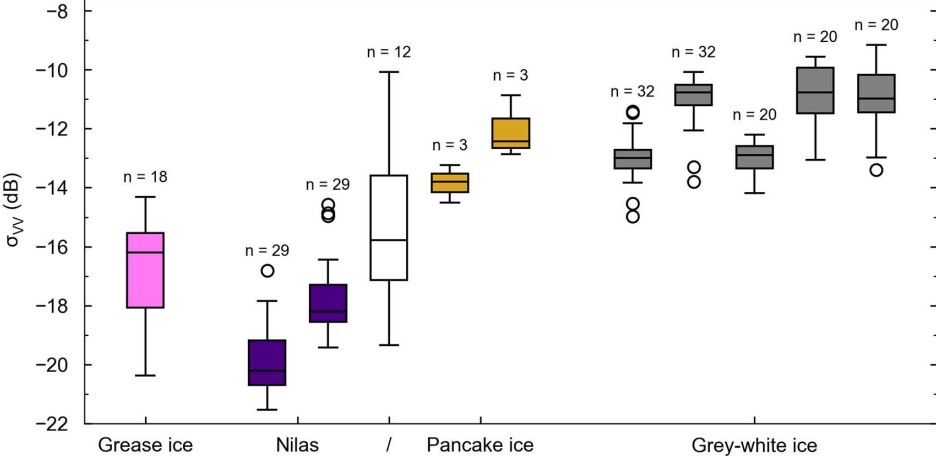

**Figure 10:** Boxplot for TerraSAR-X median VV backscattering values observed over AOIs of each ice type in 2016 and 2017. Grease ice (pink) was observed on the orbit 13 image from 26 November 2017. Nilas (dark purple) was observed on 28 and 29 November 2016 in orbits 13 and 21, respectively. A mix of nilas and pancake ice (white) was observed on 26 November 2017

in orbit 13. Pancake ice (yellow) was observed on 28 and 29 November 2016 in orbits 13 and 21. Grey-white ice (grey) was observed on 9 and 10 December 2016 in orbits 13 and 21, as well as on 1, 7 and 8 December 2017 in orbits 89, 13, and 21. The number of median values used (n) is written above each box. Outliers are plotted as empty white circles.



Figure 11 shows median backscattering time-series for the freeze-up period. Also shown are the first freezing degree-day, the day

of freeze-up, the freeze-up peak, and the beginning of monotone backscattering. These events were documented using temperature measurements, time-lapse photography, and TerraSAR-X images. Temperatures first dropped below 0°C around the same time each year: 3 October 2015, 7 October 2016, and 7 October 2017. Respectively 161, 191, and 231 freezing degree-days had accumulated at the time of freeze-up for the three years. 2015 saw both the earliest freezing degree-day and the earliest freeze-up. These two events were closer to each other in 2015; they were separated by 39 days compared to 53 and 52 days in 2016 and 2017.

Mean freezing degree-day for that period was essentially the same for all three years, sitting between 3.5 and 4.5°C. No TerraSAR-X data is available during freeze-up 2015.

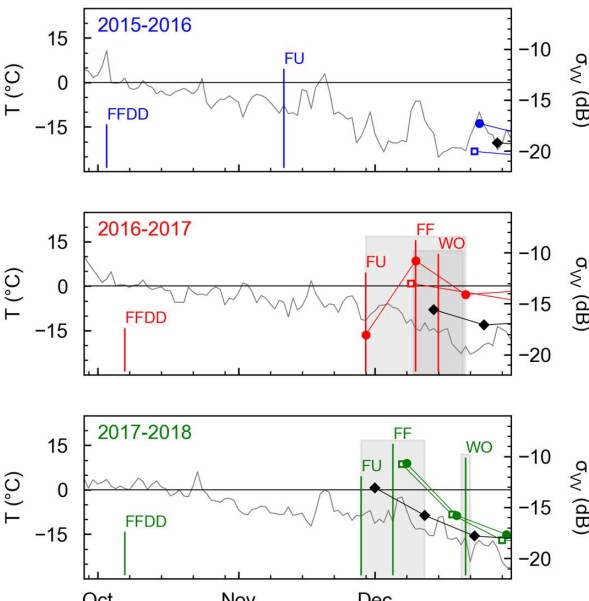

**Figure 11:** TerraSAR-X median VV backscattering (color, right) and daily mean temperature (grey, left) are plotted versus time

for each year (color-coded). Three orbits are shown for: orbits 13 (empty square), 21 (circle) and 89 (black diamond). First freezing degree-day (FFDD, from temperature), freeze-up (FU, from time-lapse photography), frost flower maximum (FF, from TerraSAR-X), and winter onset (WO, from TerraSAR-X) are identified with vertical bars. Ranges for indicators derived from TerraSAR-X are indicated by shaded grey areas. The horizontal black line indicates 0°C.

**6.2 Winter**

Median monthly temperature from September to June is presented in Table 2 for the three years of the study. The coldest months were observed in 2017-2018, with median January and February temperatures sitting at -25 and -30°C, respectively. The mildest winter was 2016-2017, with only one month featuring median temperature below -20°C.





**Table 2:** Median monthly temperature in degrees Celsius from September to June for the three years of the study. Months where the median was below 0°C are filled in blue, and median values below -20°C appear in bold.

|  | Sep. | Oct. | Nov. | Dec. | Jan. | Feb. | Mar. | Apr. | May | Jun. |
|---|---|---|---|---|---|---|---|---|---|---|
| 2015-2016 | 3 | -2 | -9 | -19 | **-22** | **-25** | **-21** | -13 | -3 | 2 |
| 2016-2017 | 7 | -1 | -5 | -15 | -18 | **-24** | -18 | -12 | -1 | 4 |
| 2017-2018 | 6 | 0 | -8 | -15 | **-25** | **-30** | -14 | -11 | -6 | 2 |

For the purpose of characterizing the winter backscattering signature of snow-covered sea ice, winter is defined from the TerraSAR-X time-series as the monotone period between the post-freeze-up peak and the spring peak. Derivation of these limits is presented in the supplementary materials (Fig. S5). The estimation range for each indicator was reduced by combining estimates from different orbit time-series. Median temperatures during the monotone backscattering period were respectively -20 ± 6 °C, -15 ± 8 °C, and -15 ± 10 °C for the three years.

The X-band winter backscattering signature of snow-covered sea ice in Deception Bay, or that of "white ice" in WMO terminology, is presented in Fig. 12. Median backscattering observed for white ice ranged from -14 to -20 dB over the three years. In winter 2015-2016, the median was consistently lower than for the other two years, across orbits. Winter values were systematically higher for the descending/morning orbit than for the ascending/evening ones, but this effect is most pronounced for 2016-2017. This is also the only year for which the descending/morning data shows a much larger spread than in the other orbits.

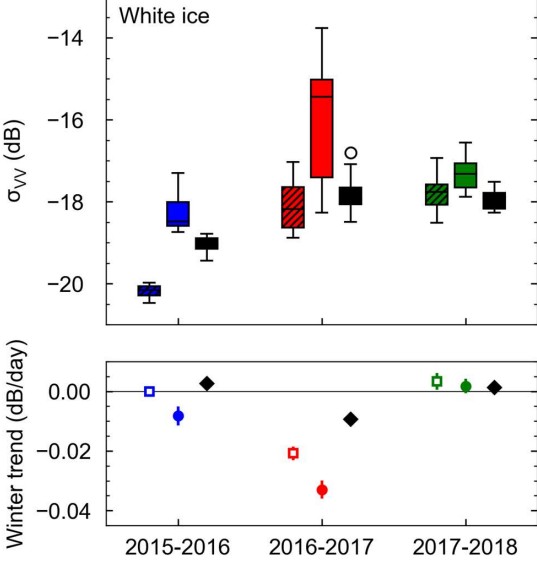

**Figure 12:** Characterization of TerraSAR-X VV winter backscattering. Top: Winter median by year (color-coded) for orbits 13 (dashed, ascending, 5:32 PM, 38°), 21 (solid, descending, 6:25 AM, 40°), and 89 (black, ascending, 5:40 PM, 46°). This seasonal median is computed from image medians, which were computed from AOI medians. Outliers are represented with an empty circle marker. Bottom: Winter trend by year (color-coded), for orbits 13 (empty square), 21 (circle) and 89 (black diamond). The zero is identified with a horizontal black line. The trend is defined as the slope of the linear fit to the winter image medians (see Fig. S6). Error bars are the standard error associated with the fit.



Winter backscattering is typically characterized by a negative trend associated with decreasing volume scattering as brine inclusions shrink due to colder temperatures (see Sect. 2.2). The trend is defined as the slope of the linear fit to the winter image medians, presented in the supplementary materials (Fig. S6). In 2016-2017, all orbits show a trend (Fig. 12). Meanwhile, 2015-2016 exhibits little to no trend, and no trend is observed in all three orbits for 2017-2018.

### 6.3 Spring

Breakup was documented using time-lapse photography and TerraSAR-X images. The daily event sequence (Tables S4-S6)—including ice-breaker transits—as well as videos assembled from time-lapse photography (Movies S3-S6), are both available as supplementary materials. In the following description of each year's breakup, we refer to zones represented in Fig. 13. The breakup date is the first day when the bay is ice-free.

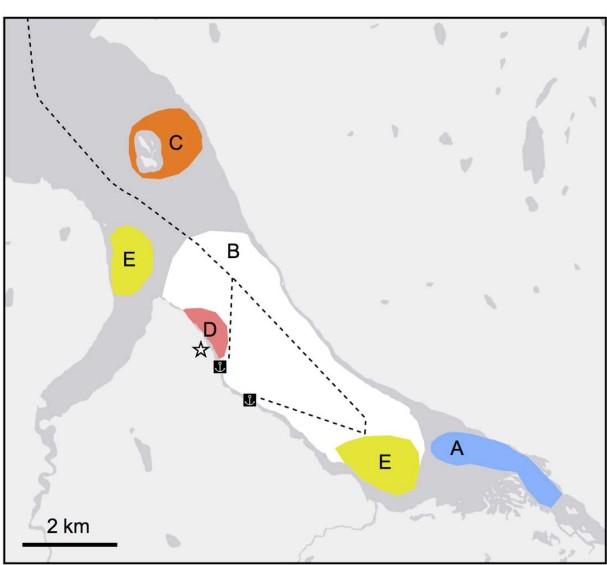

**Figure 13:** Zones relevant for describing the spatial aspects of breakup in Deception Bay, and ship routes for the MV *Arctic* and MV *Nunavik* (dashed line). Camera location is indicated with a star.

In 2016, patches of bare ice could be observed throughout the winter, particularly along the south-west shore (zone D). This bare ice started to appear rougher on 20 May, as shown in Fig. 14. Despite ice-breaking manoeuvers performed by the MV *Nunavik* in zone B upon its arrival in the bay on 16 June, no open water could be seen along its tracks either on the photographs or on the TerraSAR-X image from the same day. Deception River thawed by 16 June. Zone D was seen to be covered in meltwater on 18 June (Fig. 14c), and open water was first observed on 19 June, in front of the river (zone A). Open water progressed steadily throughout zone B over the course of five days, until Moosehead Island (zone C) was also ice-free and breakup was completed on 24 June 2016.




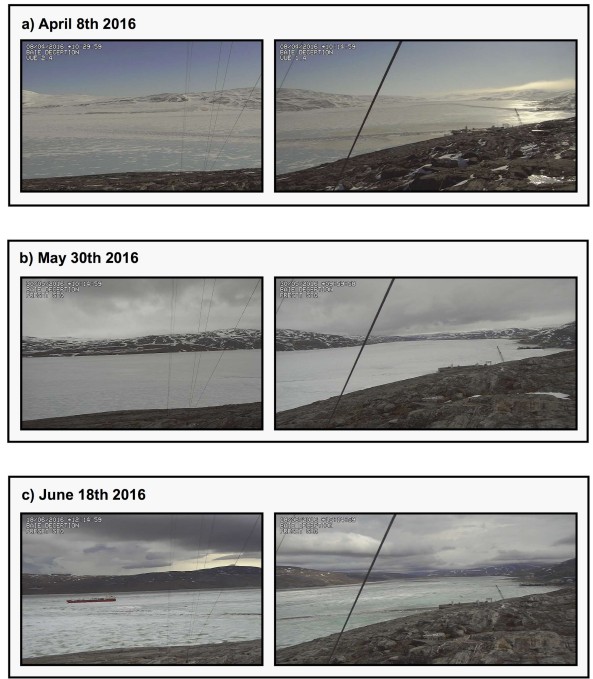

**Figure 14:** Spring 2016 as seen from time-lapse photography. a) Bare ice with a smooth appearance b) Rougher-looking ice c) Meltwater ponds appeared on the ice on the day before open water first appears in 2016. The left-hand photographs are associated with view 1, and the right-hand photographs with view 2.

In 2017, snow rapidly melted off following the end of the monotone backscattering period. By 13 May, more than two thirds of zone B was snow-free, before a snowfall event on the 14 May. On the 31 May, the ice was covered in meltwater ponds. Deception River had thawed by 3 June (zone A), and on 4 June some open water could be seen along the ship tracks near zone D. Breakup took eight days and followed the same spatial pattern as the year before. Breakup was complete with the freeing of zone C on 12 June 2017. In 2018, the snow cover appeared largely melted on the south-eastern part of zone B by 28 May, and the ice was seen to be covered in meltwater on several occasions mid-June (zones E). The MV *Nunavik* and MV *Arctic* entered the bay on 17 June. Six days later, open water could be seen along most of the ship tracks, and the river had thawed. The ships' departure coincided with the first day where the ice was covered in meltwater ponds. New cracks perpendicular to the shore appeared in the ice that day. These features can be seen on photographs reproduced in Fig. 15. Open water was first observed near the south-east shore in zone B on 26 June. The TerraSAR-X image acquired that day (Fig. 15) shows large ice pieces separated along the ship tracks and floating freely in zone B. The breakup was completed on 3 July 2018, seven days after the first observation of open water.

Figure 16 shows median backscattering time-series for the spring period. Also shown are the first thawing degree-day, the end of monotone backscattering, the spring peak, and the breakup. These events were documented using temperature measurements, TerraSAR-X images, and time-lapse photography. For each year, temperatures first increased past 0°C on 26 April 2016, 8 May 2017, and 29 May 2018, and respectively 74, 110 and 105 thawing degree-days had accumulated at the time of breakup. 2016 saw both the earliest end of monotone backscattering and the longest period between this and breakup—59 days compared to 35 both in 2017 and 2018. Mean thawing degree-day during this period was 1°C in 2016, compared to 3°C in 2017 and 2018.



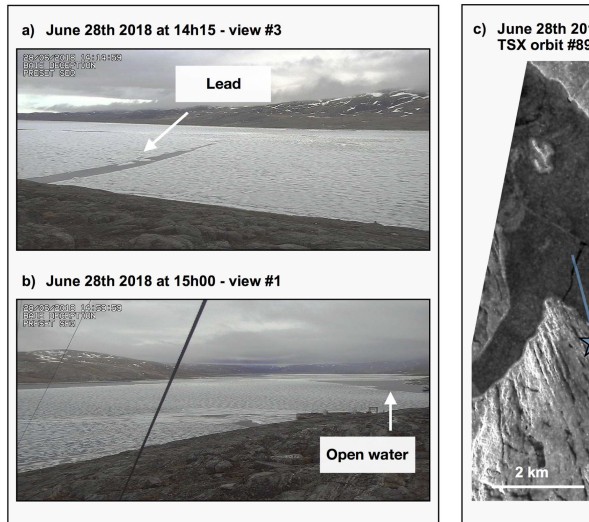

**Figure 15:** Open water and lead observed in Deception Bay on June 28th 2018. a) and b) Photographs taken from the shore showing meltwater ponds covering the ice, as well as a lead in view 3 and open water in view 1. c) TerraSAR-X VV image from orbit 89 (scaled from -19 to -5 dB). The location of the camera is identified with a blue star and the fields of view limits indicated with blue lines.

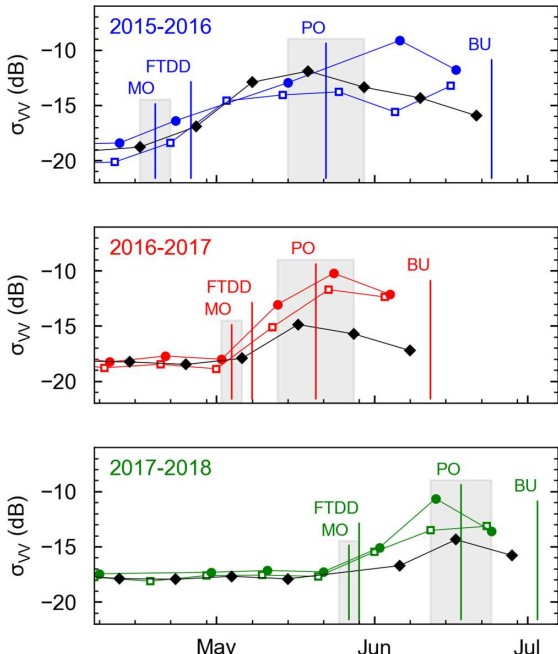

**Figure 16:** TerraSAR-X median VV backscattering is plotted versus time for each year (color-coded). Three orbits are shown: orbits 13 (empty square), 21 (circle) and 89 (black diamond). Melt onset (MO, from TerraSAR-X), first treezing degree-day (FTDD, from temperature), pond onset (PO, from TerraSAR-X), and breakup (BU, from time-lapse photography), are identified with vertical bars. Range for indicators derived from TerraSAR-X are indicated by shaded grey areas.



## 7. Discussion

### 7.1 Complementary tools for seasonal monitoring

The project's goal of monitoring snow-covered first-year sea ice in Deception Bay proved to be well-served by the combined use of time-lapse photography and radar remote sensing. Photographs were used to identify newly formed ice types (Fig. 9), which allowed us to document their X-band backscattering signature (discussed in Sect. 7.2). Conversely, springtime transitions like melt onset or water drainage from the snowpack and ponding, associated, respectively, with the end of monotone backscattering (Barber and Nghiem, 1999) and to the spring peak (Barber et al., 1995), couldn't be resolved on time-lapse photography, but were detected

from the TerraSAR-X time-series (Fig. 16). Table 3 summarizes the seasonal timeline events observed for each year of the study and the relevant data source. The following section further discusses how the TerraSAR-X specific events presented in Table 3 are linked to physical processes such as the presence of frost flowers, snow accumulation, snowmelt and water drainage from the snow. These indicators could be used as proxies for interannual monitoring, for instance of snowmelt timing and length.

**Table 3:** Seasonal timeline for snow-covered sea ice for three years, derived from three data sources: temperature (Temp.), time-lapse photography (Photo.), and TerraSAR-X (TSX).

|  | 2015-2016 | 2016-2017 | 2017-2018 | Temp. | Photo. | TSX |
|---|---|---|---|---|---|---|
| First freezing degree-day | Sep. 24th | Oct. 6th | Oct. 4th | x |  |  |
| Freeze-up | Nov. 11th | Nov. 29th | Nov. 28th |  | x |  |
| Frost flower maximum | - | Dec. 10th | Dec. 5th |  |  | x |
| Winter onset | - | Dec. 15th | Dec. 21st |  |  | x |
| Melt onset | Apr. 19th | May 4th | May 27th |  |  | x |
| First thawing degree-day | Apr. 26th | May 8th | May 29th | x |  |  |
| Pond onset | May 22nd | May 20th | Jun. 18th |  |  | x |
| First observation of open water | Jun. 19th | Jun. 5th | Jun. 26th |  | x |  |
| Break up | Jun. 24th | Jun. 12th | Jul. 3rd |  | x |  |

### 7.2 Interpretation for the X-band backscattering seasonal time-series

The TerraSAR-X backscattering time-series presented in this article exhibits the same seasonal evolution as that of the C-band (Sect. 2), which was expected due to the spectral proximity of both bands. The post-freeze-up peak is attributed to frost flower development (Isleifson et al., 2014) and decimation or masking by snow infiltration (ibid., Onstott, 1992). It was observed in full for one data series (orbit 21 in 2016-2017) and partially for all orbits in 2017-2018, as shown in Fig. 11. By combining different orbits, frost flower maximum was shown to have occurred within two weeks of freeze-up 2017 (ibid.). Winter onset—the transition

to stable and relatively low backscattering—is brought on by cold temperatures in the snow and at the ice surface, themselves due to insulation by the ice cover (Yackel et al., 2007). It was similarly observed for orbit 21 in 2016-2017 and all orbits in 2017-2018. The combination of all orbits reduced the margin of error to one day for winter onset in 2017-2018, which occurred 23 days after freeze-up. Melt onset, where meltwater begins to accumulate at the top of the snowpack and the backscattering starts to increase



(Barber and Nghiem, 1999), was detected in all of the nine TerraSAR-X time-series (Fig. 16). By combining three orbits, the
margin of error on melt onset detection was reduced to two or three days depending on the year (ibid.). Melt onset systematically
occurred before the first thawing degree-day; it may be more closely associated with the first day where temperatures increase past
zero even for only a short period of time, without the daily mean crossing the 0°C threshold. Pond onset, where water drains from
the snowpack and starts to accumulate on the ice surface, yielding a decrease in the backscattering (Yackel et al., 2007), was also
observed in all nine datasets (Fig. 16). The margin of error for this indicator was reduced to six and 7.5 days, depending on the
year, by combining three orbits. As described in the methods (Sect. 5.3), detecting a peak requires three data points, while detecting
an inflexion point only requires two—it is therefore expected that the margin of error for peak detection be double that of inflexion
point detection. Pond onset detected with TerraSAR-X preceded meltwater observation from time-lapse photography by roughly
a month in 2016 and 2017, but the two were coincident in 2018.

### 7.3 Freeze-up and breakup processes

Two different freeze-up processes were documented over three years. In 2015, calm waters allowed for a quick thermal freeze-up
which yielded smooth ice, as seen on time-lapse photography (Fig. 14). The smoothness of this ice cover contrasts with the
relatively rougher ice produced by the iterative freeze-up of 2016 and 2017 which proceeded from patches of nilas and ice rind,
and yielded higher winter backscattering (Fig. 12). Ice roughness was therefore correlated with winter backscattering values despite
the presence of a snow cover and presumably, according to Barber and Nghiem (1999) and others (see Sect. 2.2), of an associated
brine-wetted snow layer.

Two different breakup patterns were documented, potentially specific to the context of ice-breaking transport. In the first two years,
open water progressed from the thawed Deception River, its low albedo and warmer water favoring melt at its edge, until the whole
bay became ice-free. In 2018 however, the ice cover dislocated along the ship route, and the resulting pieces were eventually
cleared out of the bay (Fig. 15). Here we explore the influence of shipping on the breakup process. In 2016, ice-breaking transits
and manoeuvers left no open water in the ship's wake (Table S4). TerraSAR-X images rather showed tracks filled with broken ice
debris, indicating that the ice was already decaying. Spring was early in 2016: indeed, the spring peak occurred more than three
weeks before the first ship transit (Fig. 16). In 2017, the spring peak similarly occurred before the first ice-breaking transit of the
season, and open water had already been observed along the ship tracks event before the first summer transit (Table S5). Spring
was much later in 2018 than in the two previous years however—both the first thawing degree-day and the spring peak were almost
a month later than in 2016 (Table 3). When shipping first resumed in 2018, at the same time as the spring peak, the ice cover was
likely still relatively thick compared to the other years. The ships left open water in their wake, rather than ice debris, and their
departure a week later broke the ice cover neatly along their route (Table S6). Open water precipitated melt along the fractures,
leading to a different breakup pattern than in 2016 and 2017.

### 7.4 New values for X-band backscattering over first-year sea ice

Here we compare our X-band backscattering observations of newly formed ice types with values published in the literature for the
same frequency. The observations reported in this article of -16 ± 2 dB for grease ice and -19 ± 2 dB for nilas (Fig. 10) are higher
than X-band VV backscattering values reported by Nakamura et al. (2005) of -22.0 ± 0.5 dB for new ice (frazil, grease ice, and
nilas) at comparable incidence angles of 39° to 44°. The same goes with regards to the value of -21 dB in the X-band and HH
polarization reported by Matsuoka et al. (2001) for snow-free thin ice (nilas, grey ice), at smaller incidence angles of 23° to 25°.
These differences may be an indicator that frost flowers were present on the surfaces we observed, since such features can
considerable increase backscatter (Isleifson et al., 2014). This remains speculative since neither Matsuoka et al. (2001) or





Nakamura et al. (2005) reported on the presence or absence of frost flowers in their studies, and no in situ observations from this study can confirm the presence of such structures. In the C-band, pancake ice was reported as yielding a higher backscattering than nilas (Alexandrov et al., 2004), which is what we observe in our X-band data: pancake ice sits at -13 ± 1 dB, and a mix of nilas and pancake ice sits between values for each single ice type, at -16 ± 3 dB. For grey-white ice, we report backscattering values of -12 ± 1 dB which compare favorably with reported values of -11.9 dB in the X-band and VV polarization for newly formed ice (nilas, grey ice, and white ice up to 50 cm thick) at an incidence angle of 28° (Johansson et al., 2017).

### 7.5 Sources of error

Temperature measurements by the time-lapse cameras are less precise than a weather station measurement, but most importantly they are prone to bias through camera-heating by the sun. Despite efforts to model and correct this bias (see Sect. 5.2), the resulting temperature series might still deviate from actual air temperature in the bay. This could yield an inaccurate time-dependence for temperature, and thus shift the timing of the first freezing and thawing degree-days as well as bias their cumulative sums. The potential impact of such errors on the interpretation of the data is very limited. Indeed, the discussion presented refers almost exclusively to the time-lapse photography and remote sensing time-series; only the timing of first thawing-degree-day relative to melt onset detected with TerraSAR-X (see Sect. 7.2) could be affected by this error source.

Another potential source of error is the combination of different TerraSAR-X orbits to reduce the estimated range for inflexion points and peaks (see Sect. 5.3). This method relies on the assumption that, when it comes to these features, the backscattering behaves in the same way whether it is acquired in morning/descending or evening/ascending orbits. For example, data from all three orbits appears coherent for spring 2017 (Fig. 16), showing the same time evolution despite differences in amplitude which may be associated with incidence angle and acquisition time. On the other hand, spring 2017 (ibid) is an example of how data from one orbit, in this case morning/descending orbit 21, may behave differently than the others. In this case, disparity is mitigated by having three complementary orbits.

### 8. Conclusion

This article presented a locally-relevant case study for first-year sea ice monitoring using a combination of TerraSAR-X and time-lapse photography time-series. The seasonal evolution of X-band backscattering from first-year sea ice was shown to reproduce that of the C-band, both in ascending and descending orbits, and for incidence angles ranging from 38° and 46°. Freeze-up and breakup processes were described for Deception Bay, an area at the confluence of climate change, land use by local Inuit, and industrial pressure in the form of ice-breaking transport. Finally, new values for X-band backscattering from newly formed sea ice types were reported. We hope that the data presented here will prove useful both on a local scale to governments and industries, by serving to document important processes in the context of a changing climate and on-going development, and on a regional scale to researchers studying snow-covered sea ice. Future work in the Ice Monitoring project will build on this characterization of seasonal processes and focus on spatial variations within the bay and comparison with similar fjords, namely Salluit and Kangiqsujuaq.



**Code and data availability:** The complete time-lapse photography database can be accessed at http://caiman.ete.inrs.ca (Bernier et al., 2017). The daily mean temperature data extracted from photograph metadata, along with the corrected daily mean and nearest

Environment Canada station data, are available for both Deception Bay and Quaqtaq at https://doi.pangaea.de/10.1594/PANGAEA.905051 and https://doi.pangaea.de/10.1594/PANGAEA.905052 (Dufour-Beauséjour et al. 2019). Quicklooks for the TerraSAR-X images are available on https://doi.pangaea.de/10.1594/PANGAEA.905246 (ibid.). The code used to compute pixel statistics from the TerraSAR-X images on areas of interest is available at https://github.com/sdufourbeausejour/tiffstats (Dufour-Beauséjour, 2019).


**Video supplement:** Movies S1, S2, and S3 respectively show the freeze-up sequence for 2015, 2016, and 2017 (Dufour-Beauséjour et al. 2019). Movies S4, S5, and S6 respectively show the breakup sequence for 2016, 2017, and 2018 (ibid.). All movies are available online at https://doi.pangaea.de/10.1594/PANGAEA.904956.

**Author contributions:** SDB participated in study design, data acquisition, analysis, and interpretation, and wrote the article. AW participated in study design, data analysis and interpretation, and wrote the article. YG participated in study design and data interpretation, and revised the article. MB participated in study design and revised the article. JP and VG participated in study design and data acquisition. JT and A. Roth participated in data acquisition. A. Rouleau participated in study design.

**Competing interests:** The authors declare that they have no conflict of interest.

**Acknowledgements:** The authors would like to thank the Inuit guides from Salluit who participated in data acquisition in Deception Bay (in alphabetical order): Chris Alaku, Johnny Ashevak, Michael Camera, Putulik Cameron, Charlie Ikey, Luuku Isaac, Markusi Jaaka, Adamie Raly Kadjulik, Joannasie Kakayuk, Jani Kenuajuak, Pierre Lebreux, Casey Mark, Denis Napartuk,

Eyetsiaq Papigatuk, and Kululak Tayara. Thanks also to INRS students who also participated in data acquisition: Pierre-Olivier Carreau, Étienne Lauzier-Hudon, and Valérie Plante Lévesque. Thanks to Jasmin Gill-Fortin (INRS) for his help with the time-lapse photography data. We further thank Charles Gignac (INRS), Randy Scharien (University of Victoria), Torsten Geldsetzer (Natural Resources Canada), and Derek Mueller (Carleton University) for their advice and suggestions on this manuscript, and Guillaume Légaré (INRS) for his advice on the temperature time-series analysis. Thanks to the German Space Agency (DLR) for

providing the TerraSAR-X images, and for data processing with the MultiSAR-System. The authors acknowledge the use of TerraSAR-X (ⓒDLR 2017-18). This study was done within the Ice Monitoring project, a research collaboration between the Kativik Regional Government (KRG), Raglan Mine, a Glencore company, Institut national de la recherche scientifique (INRS) and the Northern Villages of Salluit and Kangiqsujuaq.



**Financial support:** This INRS research was supported by Polar Knowledge Canada (Safe Passage, project number PKC-NST-1617-0003), Raglan Mine (a Glencore company), the Kativik Regional Government, the NSERC Discovery Grant -and the Northern Research Supplements Program (attributed to Pr. Monique Bernier), the Ministère des Transports du Québec, and the Northern Scientific Training Program (attributed to Sophie Dufour-Beauséjour). Ph.D. scholarships were provided to the first author by NSERC (Alexander Graham Bell Canada Graduate Scholarship – Doctoral) and the W. Garfield Weston Foundation (The W. Garfield Weston Awards for Northern Research).

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
