# Peer review of "Seasonal timeline for snow-covered sea ice processes in Nunavik's Deception Bay from TerraSAR-X and time-lapse photography"

_The Cryosphere, 2019_

## Referee Comment (RC1) · Anonymous Referee #1 · 30 Oct 2019

This paper analyses the seasonal evolution of X-band VV backscatter from snow-covered first-year sea ice from Deception Bay, using TerraSAR-X with collocated time-lapse photography, with focus on freeze-up and break-up stage characterization. This is a timely research topic especially with various regions in the Canadian Arctic experiencing greater variability in seasonal sea ice conditions, inter-annually and that SAR is a useful tool to monitor these changes. The authors also need to appreciate their effort to collect and process a lot of time-lapse photography and make good use of it.

However, with the current draft, this manuscript needs considerable discussion based on the observations (currently, there are way too many assumptions, especially the

scattering mechanisms during the seasonal regime). The paper, although focuses importantly on a community-level study, suffer from several shortcomings that weaken significantly its message and needs to be addressed before the paper could be accepted. I list my major concerns now. I have left out the minor comments for now and will review them in the revised version.

Major Comments.

My major concern with this paper now is how authors have justified the similarity in the backscatter evolution of X-band and C-band. See Line 485 under section 7.2. "The TerraSAR-X backscattering time-series presented in this article exhibits the same seasonal evolution as that of the C-band (Sect. 2), which was expected due to the spectral proximity of both bands.". This sentence reads like the author already knew about the results and as an afterthought. This has lead to authors more or less assuming the scattering mechanisms during the seasonal evolution (like that with C-band), based on past literature. This is scientifically misleading. If there was similarity in scattering mechanisms at two different frequencies, our scientific community wouldn't have launched TerraSAR-X and RADARSAT-2 (for e.g.).

Although the reviewer agrees with the observations from the time-lapse photography related to freeze-up and break-up processes, the authors provide little to no information about

a) Although the objective of this manuscript was to focus more on how X-band SAR can be used to provide the first-baseline signature of X-band VV backscatter. However, the majority of the paper is about analyses from time-lapse photographs and very little focus was given to analyzing the SAR signature section. I would suggest using the SAR images as the focal point of analysis (with snow/sea-ice geophysical explanation of changes in VV backscatter), 'supported' by time-lapse photography.

a) how they classified ice types (what method) from the TerraSAR-X images, based on beta-naught values? What is the advantage of using beta-naught over traditional

sigma-naught? The authors may be reminded that the scattering mechanisms discussed in this paper (mostly based on previous literature) are applicable for sigma-naught values (significantly dependent on polarization). Therefore, substantial justification should be provided on why beta-naught values are used. And if they are, how does the scattering mechanisms change?

b) The interesting part is how authors easily interpret different ice types (grease ice, nilas, pancake ice, and grey-white ice) without any geophysical explanation (or the least scattering mechanism) justifying the backscatter occurrence from these ice types. This needs to be clarified. Although the authors have demonstrated diversity in VV (figure 10) for different ice types, the authors should demonstrate the proof of how they classified or interpreted them as these 'specific' ice types. For another example, the authors talk about 'frost flower maximum' which causes the first X-band inflection point. But the authors do not provide any proof of frost flower formation

c) The third missing point of this paper is the lack of scattering mechanism explanation (mostly assumptions and backing up from past literature on C-band now) or sometimes explaining without any clarity in this regard. The authors should explain what they observe from the VV backscatter, based on the incidence angle range used in this study (and if they have in situ observations of snow and sea ice properties) and NOT based on agreeing with that they see from the SAR imagery, against past literature (using different incidence angle ranges from C-band imagery).

d) If the authors haven't noticed, one advantage of the X-band signature time series across three years is its utility to detect melt and pond onset from SAR images (which is always challenging) and how varied the dates are for these three years. The authors, if interested should consider using this application as a tool to improve this manuscript. In addition to freeze-up and break up, another application in which the science community and also local communities are interested in how the timing of melt and ponding changes and how it can be effectively detected from SAR images. Just a suggestion for improvement.

Overall, if the authors would like to stick with the objective to provide a baseline understanding of X-band signature evolution, here are my suggestions

a) Even though data for all three years are available, use signatures from one year as the baseline and study the evolution of the X-band signature. That would be your baseline (which should also include describing the X-band scattering mechanisms).

b) With lack of in situ snow and sea ice observations of geophysical properties, the authors have the freedom to speculate the scattering mechanisms (never a drawback, and always room for improvements) instead of blind conviction.

c) Once the baseline signature is explained for one season, use it to differentiate different core regimes changes in the region. For eg. Table 3 shows differences in winter onset, melt onset and pond onset from SAR images for all three years. Use this info as a strong point to showcase the utility of X-band to effectively detect these changes (which can be then integrated into talking about the importance for local communities).

d) Use time-lapse photographs more as an ancillary data to explain the X-band signature evolution, and not the other way. Remember what your primary objective is.

---

## Referee Comment (RC2) · Anonymous Referee #2 · 6 Nov 2019

The manuscript presents monitoring of the seasonal evolution of sea ice cover in a Canadian fjord, using satellite images and photographs. They use TerraSAR-X images and co-located photographs to monitor the fjord to identify different development stages of the sea ice. The manuscript is within the scope of the journal and though the manuscript contains some interesting results it needs to be considerably rewritten before it can be accepted.

Specific comments The abstract is rather imprecise, e.g. it is claimed that Inuit's have reported greater inter-annual variability in the seasonal ice conditions. In which way were there changes? Since when have they reported this? This information is very

useful and it would have been very nice if these observations were further reported and explored within the manuscript. Why can we expect increase in solid precipitation? Over which time period? Please rewrite the abstract to focus on the main findings and points addressed within the manuscript.

The manuscript is very long and contain information that is well covered in other works, e.g. the sea ice evolution during the year. Please reference these works instead, and only highlight things of specific importance and relevant to the scientific work carried out within this manuscript. This would significantly shorten the manuscript, e.g. can section 2 be significantly shortened to possibly cover $\frac{1}{2}$ page instead of the near 3 pages. The study area section can also be shortened, e.g. is the tidal range not important for the rest of the study. Similarly, is the last paragraph in section 3 not relevant for the presented work. Please revise the work bearing in mind what you are trying to convey and new scientific findings.

Please expand the methods section to explain to the reader what is done in the study and how the results are achieved. E.g. define and explain why the following is calculated; first freezing degree-day, freeze-up, frost flower maximum and winter onset?

According to the manuscript frost flowers could not be observed in the photographs, and as far as the reader can work out a peak in SAR backscatter values is therefore inferred to correspond to frost flowers. Though this is not again specifically stated in the methods section. Moreover, how do you know that frost flowers were present? Could the post-freeze-up peak be related to increased sea ice thickness?

Why are not snow thickness information reported for any other time period than April 2017? In order to fully address the scientific topic indicated in the title of the paper information about the snow thickness is essential for the sea ice monitoring. It is unclear to the reader how are the estimates about the snow cover carried out? Photographs? Is there information about snow cover thickness and distribution? Judging by the title the manuscript should only contain information about snow-covered sea ice. Yet figures

showing e.g., grease ice and pancake ice has no snow cover. Consider updating the title to reflect the sea ice included within the study. Please discus how do you expect the snow-cover to affect the results?

Why is beta nought used instead of the more commonly used sigma nought? Since beta nought is used please use the symbol beta in all the figures 6, 10, 11, 12 and 16 instead of the symbol sigma.

The incidence angle difference within the dataset is significant (8o), how does this affect the presented results? The fjord is given as 20 km wide (a scale bar would be nice to see in Figure 2), how much of the overall area is covered by the 15km wide TerraSAR-X images?

It is in the discussion stated that the sea ice observed in the ship wake was broken. How is this verified? How are wind effects accounted for in these observations? Would it be possible to include observations from these ships?

It is in the manuscript stated that the X-band backscatter change is expected to be similar to one in C-band. A suggested to corroborate this is to include Sentinel-1 images overlapping the fjord to investigate if these C-band images show the same evolution as the TerraSAR-X images. The use of Sentinel-1 may also reduce the revisit time.

Rather than stating that the standard meteorological station at the airport is not used, a correlation measure between the used temperature and the airport temperature would have been beneficial. Such a comparison would also have verified the supposedly heated camera case claim, rather than a statement that the authors think that it is so. At what altitude is the temperature sensor located?

As stated in the manuscript is FDD used, yet the unit used was oC, please explain? Also freezing at +3.5 oC to +4oC seems a bit high. Why is 0oC used when sea ice is investigated as it will likely freeze at -1.8oC. Are there any sea water temperature measurements? It is mentioned in the discussion that the Deception river has warm

water, please provide a temperature time series for this river or at least give some specific temperatures.

How do the values in e.g. Fig. 10 relate to e.g. work by Onstott 1992?

Additional comments According to the temperature records, as presented on row 254, temperatures are recorded between 11 September 2015 – 16 September 2016 and then from 18 September 2018 (?) to 31 August 2018. Please clarify.

Ibid. is not a way to reference that I've seen in this journal before. Whilst this might still be ok, many of the references where ibid. has been used are not correct and the statements that are supposed to be covered in those references are not included here. E.g. on row 129 the "white ice" term is attributed to Johansson et. al., 2017 and on row 400 correctly to WMO. Moreover, when referring to the WMO terminology please include the full reference, (WMO, 2014).

---

## Author Comment (AC1) · 20 Dec 2019

Dear **Referee #1**,

Thank you for your insightful comments on the manuscript and for providing advice on how to improve it. We appreciate your time. The manuscript has been considerably reworked following your comments and those of Referee #2.

The title and objectives have been reworded to reflect our focus on the combined use of TerraSAR-X and time-lapse photography for seasonal sea ice processes monitoring. Section 2 "SAR backscattering from sea ice" has been removed. The methods and results have been re-organized and some content has been moved to the supplementary materials. The discussion has been completely rewritten.

We reproduced your comments below (R), provided our answers (A), and detailed changes to the manuscript (M). When providing section numbers, we refer to the first version of the manuscript.

SDB

**Anonymous Referee #1**

**R1:** My major concern with this paper now is how authors have justified the similarity in the backscatter evolution of X-band and C-band. See Line 485 under section 7.2. "The TerraSAR-X backscattering time-series presented in this article exhibits the same seasonal evolution as that of the C-band (Sect. 2), which was expected due to the spectral proximity of both bands." This sentence reads like the author already knew about the results and as an afterthought. This has lead to authors more or less assuming the scattering mechanisms during the seasonal evolution (like that with C-band), based on past literature. This is scientifically misleading. If there was similarity in scattering mechanisms at two different frequencies, our scientific community wouldn't have launched TerraSAR-X and RADARSAT-2 (for e.g.).

**A1:** All assumptions of similarity between both bands have been removed from the manuscript. Comparison of the X-band data with the literature on C-band is now reserved for the discussion.

**M1:**

Section "2. SAR backscattering over snow-covered sea ice", which presented a literature review on the seasonal evolution of C-band backscattering from first-year sea ice, was removed following your comments as well as those of Referee #2. Relevant references to the literature on this topic are now reserved for the discussion.

In the Methods, the seasonal features consistently observed throughout the acquisition parameters and years of the study are no longer associated to physical processes or mechanisms:

> "Recurring seasonal features in all X-band VV median backscattering time-series acquired during this study include two peaks separated by a monotone period. From this, four indicators were derived: the post-freeze-up peak (I), the beginning (II) and

*end (III) of the monotone period, and the spring peak (IV). Examples are shown in Fig. 4 for two different years and orbits, chosen for their clarity.*

[Figure]

***Figure 4****: Examples of change detection in TerraSAR-X VV median backscattering. Peak detection for orbit 21 in 2016-2017 (top), and inflexion detection for orbit 13 in 2017-2018 (bottom)."*

*(in the Methods)*

In the Discussion, each seasonal feature is examined in terms of potential scattering mechanisms.

*"The post-freeze-up peak and monotone backscattering onset are also observed in C-band time-series over sea ice (Yackel et al., 2007), but these features have been less studied than their spring counterparts (end of monotone backscattering and spring peak). Moreover, the same features in the X and C-band could well be related to different scattering mechanisms, and even to different physical processes. We limit ourselves to speculating, for the X-band data presented in this manuscript, that the increasing portion of the peak may be associated with the domination of surface scattering related to a brine-rich ice surface, potentially covered in frost flower, and that the decreasing portion may be associated with a transition to a dispersion regime, in which the signal suffers loss in the brine-wetted and increasingly colder snow. " (in the Discussion)*

**R2:** a) Although the objective of this manuscript was to focus more on how X-band SAR can be used to provide the first-baseline signature of X-band VV backscatter. However, the majority of the paper is about analyses from time-lapse photographs and very little focus was given to analyzing the SAR signature section. I would suggest using the SAR images as the focal point of analysis (with snow/sea-ice geophysical explanation of changes in VV backscatter), 'supported' by time-lapse photography.

**A2:** The manuscript title and objectives were reworded to clarify the focus of the work, which is on the combined use of TerraSAR-X and time-lapse photography time-series for the seasonal monitoring of sea ice processes. Both observational tools are uniquely qualified for remote applications, for instance in polar regions, and are often used as stand-alone tools. However,

they provide access to different aspects of the environment they observe, and have different strengths (e.g. photography allows for hourly acquisitions, but with a limited view, while SAR remote sensing has a wide and precise spatial coverage, but with fewer acquisitions). We chose to give equal importance to the two data sources to explore their complementarity. The manuscript has been reworked to focus on this objective. The Methods and Discussion sections have been reorganized in the following way: first, each data source is treated as a stand-alone monitoring tool, and second, the two data sources are co-interpreted.

**M2**:

Reworded Title:

> *"Combining TerraSAR-X and time-lapse photography for seasonal sea ice monitoring: the case of Deception Bay, Nunavik" (Title)*

Reworded manuscript objectives:

> *"This article explores the use of combined TerraSAR-X and time-lapse photography time-series to observe seasonal sea ice processes, and the potential of the time-lapse photography to support TerraSAR-X interpretation. The case study is performed over three years in Nunavik's Deception Bay. A complementary objective is to describe the processes through an interannual comparison. (in the Introduction)*

The Methods have been expanded and reorganized to clarify our parallel use of photograph interpretation and TerraSAR-X image interpretation, and their co-interpretation:

> *"[...] Sections 4.1 and 4.2 describe the indicators and how they are observed or measured from each data source. Section 4.3 then explains how photographs are compared with coincident satellite images and used to identify their features, which serves to evaluate the potential of time-lapse photography to enhance TerraSAR-X image interpretation." (in the Methods)*

The Discussion has been rewritten:

> *"The use of TerraSAR-X and time-lapse photography time-series for seasonal monitoring of sea ice processes is first discussed for each data source as a stand-alone monitoring tool (Sect. 6.1), and then for their combination (Sect. 6.2). This discussion focuses on three aspects of sea ice processes which are accessible with these tools: temporal, spatial, and spectral. Section 6.3 then discusses seasonal sea ice processes observed using combined TerraSAR-X and time-lapse photography time-series." (in the Discussion)*

**R3:** b) how they classified ice types (what method) from the TerraSAR-X images, based on beta-naught values? What is the advantage of using beta-naught over traditional sigma-naught? The authors may be reminded that the scattering mechanisms discussed in this paper (mostly based on previous literature) are applicable for sigma-naught values (significantly dependent on polarization). Therefore, substantial justification should be provided on why beta-naught values are used. And if they are, how does the scattering mechanisms change?

**A3:** In the Methods section, it was incorrectly indicated that the TerraSAR-X data had been processed in beta-naught. The data is actually in the conventional sigma-naught, which is why the sigma symbol is used throughout the manuscript. Ice type classification was performed based on photograph interpretation (see R4 and answers).

**M3**: Corrected:

> *"This workflow starts with a conversion from the digital number to radar brightness (sigma-naught) [...]." (in the Methods)*

**R4:** c) The interesting part is how authors easily interpret different ice types (grease ice, nilas, pancake ice, and grey-white ice) without any geophysical explanation (or the least scattering mechanism) justifying the backscatter occurrence from these ice types. This needs to be clarified. Although the authors have demonstrated diversity in VV (figure 10) for different ice types, the authors should demonstrate the proof of how they classified or interpreted them as these 'specific' ice types.

**A4:** Ice type identification was performed based on photograph interpretation, by following the WMO nomenclature (WMO, 2014). Ice type backscattering signature was extracted by co-interpreting the photographs and satellite images. This has been clarified in the Methods. Specifically, grease ice, nilas, and pancake ice types were observed on the photographs. Since grey-white ice is essentially characterized by its thickness, we removed the identification of this type of ice and instead refer to ice less than two weeks old as "unidentified ice".

**M4**:

In the Methods, a section is reserved to describe remote sensing and photograph co-interpretation, with examples:

> *"TerraSAR-X images were interpreted spatially using coincident photographs taken from the shore. Observed features include open water areas or leads and different ice types. Figure 6 shows two examples. At the top, nilas, pancake ice and grease ice are observed on the photographs during the 2017 freeze-up process, and then identified on a coincident TerraSAR-X image from 26 November.*

[Figure]

[Figure]

*Figure 6: Coincident time-lapse photography and TerraSAR-X image during the 2017 freeze-up process. On the image, camera location and fields of view are identified in blue. The TerraSAR-X VV image, grey-scaled from -19 to -5 dB, is from orbit 13. AOIs are color-coded according to the identified ice type, prior to backscattering signature extraction." (in the Methods)*

In the Results, backscattering data is presented for ice types identified from photographs and for unidentified young ice less than two weeks old:

[Figure]

[Figure]

*Figure 8: TerraSAR-X median VV backscattering values observed over AOIs of ice types identified from time-lapse photography in 2016 and 2017. The number of median values used (n) is written above each box. Outliers are plotted as empty white circles. Left: Grease ice (pink) was observed on the orbit 13 image from 26 November 2017. Nilas (dark purple) was observed on 28 and 29 November 2016 in orbits 13 and 21,*

*respectively. A mix of nilas and pancake ice (white) was observed on 26 November 2017 in orbit 13. Pancake ice (yellow) was observed on 28 and 29 November 2016 in orbits 13 and 21. Right: Unidentified young ice (grey) was observed on 9 and 10 December 2016 in orbits 13 and 21, as well as on 1, 7 and 8 December 2017 in orbits 89, 13, and 21. The number of days since the freeze-up date (t) is written below each box. (in the Results)*

**R5**: For another example, the authors talk about 'frost flower maximum' which causes the first X-band inflection point. But the authors do not provide any proof of frost flower formation.

**A5**: We agree that the post-freeze-up peak cannot be reliably attributed to the presence of frost flowers. Indeed, frost flowers are too small to be resolved on the photographs.

**M5**: As described in our answer to R1 (above), association of seasonal features (e.g. post-freeze-up peak) to physical processes and scattering mechanisms has been removed from the Methods and Results, and is instead reserved for the Discussion, when possible.

**R5:** d) The third missing point of this paper is the lack of scattering mechanism explanation (mostly assumptions and backing up from past literature on C-band now) or sometimes explaining without any clarity in this regard. The authors should explain what they observe from the VV backscatter, based on the incidence angle range used in this study (and if they have in situ observations of snow and sea ice properties) and NOT based on agreeing with that they see from the SAR imagery, against past literature (using different incidence angle ranges from C-band imagery).

**A5:** Following your comments and those of Referee #2, we added some discussion on the effect of the incidence angle range used in the study. In the absence of in situ observations (given the focus of this paper on the combined use of two remote observation tools), definitive explanation of the scattering mechanisms is not possible. We however provide hypotheses for the mechanisms responsible for the seasonal features, which are more or less involved depending on the available literature (e.g. it is harder to speculate on mechanisms causing the post-freeze-peak than on those associated with melting and ponding).

**M5**:

The discussion was rewritten and includes segments on the scattering mechanisms for each seasonal process (examples are M1, M6). To avoid the logical fallacies you identified in your comments (e.g. X-band = C-band, or "cause of C-band feature" = "cause of X-band feature"), they are structured as follows:

1. Description, from the results, of an X-band feature
2. Existence, from the literature, of a seasonnaly coincident similar feature in the C-band (ex. inflexion point, peak)
3. Description, from the literature, of scattering mechanisms and snow or sea ice processes explaining this C-band feature
4. Discussion, from speculation, on how these mechanisms may translate or not to the X-band, in the event of these snow or sea ice processes

Added a discussion on incidence angle:

*"Before moving on to the spring processes, we first discuss the influence of an 8°
difference between ascending orbits 13 and 89. For 2016-2017 and 2017-2018, a
small incidence angle effect was seen during the post-freeze-up and spring peaks,
where backscattering was 1 to 3 dB smaller at the higher incidence angle, and no
effect was seen during the monotone winter period (see Fig. 9 and 12). A
backscattering signal which decreases with incidence angle is expected for situations
dominated by surface scattering on a relatively rough surface (Ulaby et al., 1986). In
the C-band, surface scattering at the interfaces between dry snow, brine-wetted snow
and ice is indeed expected to dominate for cold snow-covered sea ice, with a transition
to mixed scattering for thicker snow covers (Gill et al., 2015). We speculate that
surface scattering on the ice formed from nilas patches explains the dependence on
incidence angle observed in our X-band data. 2015-2016 however presents a very
different case. Backscattering at the higher incidence angle is consistently 2 dB
stronger than at the lower incidence angle, throughout winter and during the spring
peak (see Fig. 12). We've shown the freeze-up process to have been different that
year compared to 2016 and 2017, and already suggested that the ice cover was much
smoother for the 2015-2016 season. We speculate that surface scattering was
consistently low that year, and that volume scattering, which Ulaby et al. (1986) have
shown can increase with incidence angle, dominated instead."* (in the Discussion)

**R6:** e) If the authors haven't noticed, one advantage of the X-band signature time series across
three years is its utility to detect melt and pond onset from SAR images (which is always
challenging) and how varied the dates are for these three years. The authors, if interested
should consider using this application as a tool to improve this manuscript. In addition to
freeze-up and break up, another application in which the science community and also local
communities are interested in how the timing of melt and ponding changes and how it can be
effectively detected from SAR images. Just a suggestion for improvement.
**A6:** A discussion on the mechanisms which may reasonably explain the link between spring
features (end of monotone backscattering and spring peak) and spring processes (melt onset
and pond onset) has been added.
**M6**:
The end of monotone backscattering in the X-band was explained as follows:
*"Monotone X-band backscattering was observed every winter of the study, for all
incidence angles and acquisition times, before a systematic springtime increase in
backscattering. In the C-band, monotone backscattering is also observed in the winter,
ending with melt onset brought on by warmer air temperatures (Yackel et al., 2007).
Mechanisms which may increase C-band backscattering from snow-covered sea ice
include surface scattering from the brine-wetted layer at the bottom of the snowpack
(Nandan et al., 2016), volume scattering on brine inclusions enlarged by an increase in
temperature (Barber and Nghiem, 1999), and surface scattering on wet snow (Gill et
al., 2015; Yackel et al., 2007) accumulated at the top of the snowpack due to
above-zero temperatures and solar radiation (Gogineni et al., 1992; Kim et al., 1984).
We speculate that the X-band is susceptible to all of these C-band mechanisms, with*

*an emphasis on surface scattering due to its lower penetration depth (Nandan et al., 2016), and attribute the end of X-band monotone backscattering to melt onset." (in the Discussion)*

The spring peak in the X-band was explained as follows:

*"Springtime backscattering was seen to eventually peak in all TerraSAR-X datasets (Fig. 12), although one series featured more than one maximum (orbit 13, 2015-2016), another none (orbit 13, 2017-2018), and an apparent mismatch between maximum location in the 2015-2016 data. In the C-band, springtime peaking of the backscattering is attributed (Yackel et al., 2007; Barber et al., 1995) to the transition from the pendular regime, where water is held in the snowpack (Scharien et al., 2012), to the funicular regime where meltwater drains downward (Scharien et al., 2012), flushing out brine (Barber et al., 1995), and potentially refreezing (Gogineni et al., 1992). Mechanisms which may decrease the C-band backscattering following this transition are attributed to a decrease in the dielectric properties of the snowpack following water drainage (Yackel et al., 2007). We speculate that the decrease in the X-band springtime backscattering is also caused by pond onset, and associated with increased penetration in the snowpack after water has drained out of it." (in the Discussion)*

**R7:** Overall, if the authors would like to stick with the objective to provide a baseline understanding of X-band signature evolution, here are my suggestions
a) Even though data for all three years are available, use signatures from one year as the baseline and study the evolution of the X-band signature. That would be your baseline (which should also include describing the X-band scattering mechanisms).
b) With lack of in situ snow and sea ice observations of geophysical properties, the authors have the freedom to speculate the scattering mechanisms (never a drawback, and always room for improvements) instead of blind conviction.
c) Once the baseline signature is explained for one season, use it to differentiate different core regimes changes in the region. For eg. Table 3 shows differences in winter onset, melt onset and pond onset from SAR images for all three years. Use this info as a strong point to showcase the utility of X-band to effectively detect these changes (which can be then integrated into talking about the importance for local communities).
d) Use time-lapse photographs more as an ancillary data to explain the X-band signature evolution, and not the other way. Remember what your primary objective is.

**A7**: As described in A2, we chose to focus on the objective of combining TerraSAR-X and time-lapse photography time-series for seasonal monitoring of sea ice processes. The focus is therefore now less on the seasonal evolution of the X-band signal from sea ice, but rather on sea ice process monitoring through a combination of the two data sources.

The language was adapted throughout the manuscript to remove assumptions regarding scattering mechanisms and instead provide hypothetical explanations (see examples in M5 and M6).

**M7**: See M2, M5 and M6.

**References**

Barber, D. G. and Nghiem, S. V.: The role of snow on the thermal dependence of microwave backscatter over sea ice, J. Geophys. Res.-Oceans, 104(C11), 25789–25803, https://doi.org/10.1029/1999JC900181, 1999.

Barber, D. G., Papakyriakou, T. N., Ledrew, E. F. and Shokr, M. E.: An examination of the relation between the spring period evolution of the scattering coefficient (σ) and radiative fluxes over Jandfast sea-ice, Int. J. Remote Sens., 16(17), 3343–3363, https://doi.org/10.1080/01431169508954634, 1995.

Gill, J. P. S., Yackel, J. J., Geldsetzer, T. and Fuller, M. C.: Sensitivity of C-band synthetic aperture radar polarimetric parameters to snow thickness over landfast smooth first-year sea ice, Remote Sens. Environ., 166, 34–49, https://doi.org/10.1016/j.rse.2015.06.005, 2015.

Gogineni, S. P., Moore, R. K., Grenfell, T. C., Barber, D., Digby, S. and Drinkwater, M.: The effects of freeze-up and melt processes on microwave signatures, in Microwave remote sensing of sea ice, Geophys. Monogr. vol. 68, edited by F. D. Carsey, pp. 329–341., Washington, DC, 1992.

Kim, Y.-S., Onstott, R. and Moore, R.: Effect of a snow cover on microwave backscatter from sea ice, IEEE J. Oceanic Eng., 9(5), 383–388, https://doi.org/10.1109/JOE.1984.1145649, 1984.

Nandan, V., Geldsetzer, T., Islam, T., Yackel, John. J., Gill, J. P. S., Fuller, Mark. C., Gunn, G. and Duguay, C.: Ku-, X- and C-band measured and modeled microwave backscatter from a highly saline snow cover on first-year sea ice, Remote Sens. Environ., 187, 62–75, https://doi.org/10.1016/j.rse.2016.10.004, 2016.

Scharien, R. K., Yackel, J. J., Barber, D. G., Asplin, M., Gupta, M. and Isleifson, D.: Geophysical controls on C band polarimetric backscatter from melt pond covered Arctic first-year sea ice: Assessment using high-resolution scatterometry, J. Geophys. Res.-Oceans, 117(C00G18), https://doi.org/10.1029/2011JC007353, 2012.

Ulaby, F. T., R. K. Moore, and A. K. Fung. 1986. Microwave Remote Sensing: Active and Passive. Vol. Volume 3-From theory to applications. United States: Addison-Wesley Publishing Company. https://ntrs.nasa.gov/search.jsp?R=19860041708.

WMO: Sea-Ice Nomenclature, No. 259, World Meteorological Organization, Switzerland, 2014.

Yackel, J. J., Barber, D. G., Papakyriakou, T. N. and Breneman, C.: First-year sea ice spring melt transitions in the Canadian Arctic Archipelago from time-series synthetic aperture radar data, 1992–2002, Hydrol. Process., 21(2), 253–265, https://doi.org/10.1002/hyp.6240, 2007.

---

## Author Comment (AC2) · 20 Dec 2019

Dear **Referee #2**,

Thank you for your insightful comments on manuscript tc-2019-199 "Seasonal timeline for snow-covered sea ice processes in Nunavik's Deception Bay from TerraSAR-X and time-lapse photography." and for providing advice on how to improve it. We appreciate your time. The manuscript has been considerably reworked following your comments and those of Referee #1.

The title and objectives have been reworded to reflect our focus on the combined use of TerraSAR-X and time-lapse photography for seasonal sea ice processes monitoring. Section 2. "SAR backscattering from sea ice" has been removed. The methods and results have been re-organized, and some content has been moved to the supplementary materials. The discussion has been completely rewritten.

We reproduced your comments below (R), provided our answers (A), and detailed changes to the manuscript (M). When providing section numbers, we refer to the first version of the manuscript.

Sophie Dufour-Beauséjour, for the authors
sophie.dufour-beausejour@ete.inrs.ca

**R1:** The abstract is rather imprecise, e.g. it is claimed that Inuit's have reported greater inter-annual variability in the seasonal ice conditions. In which way were there changes? Since when have they reported this? This information is very useful and it would have been very nice if these observations were further reported and explored within the manuscript. Why can we expect increase in solid precipitation? Over which time period? Please rewrite the abstract to focus on the main findings and points addressed within the manuscript.
**A1:** The abstract was rewritten to focus on the main points addressed in the manuscript.
**M1**:
We added examples of changes to seasonal sea ice conditions reported by Inuit:
 *"Indeed, Inuit have reported greater inter-annual variability in seasonal sea ice conditions, including later freeze-up and earlier breakup." (in the Abstract)*
Regarding climate projections, we clarified the information:
 *"The evolution of seasonal sea ice conditions in Deception Bay is expected to continue, with 2040-2064 climate projections for the region showing shorter snow cover periods and warmer annual average temperature (Mailhot and Chaumont, 2017)." (in the Introduction)*

**R2:** The manuscript is very long and contain information that is well covered in other works, e.g. the sea ice evolution during the year. Please reference these works instead, and only highlight things of specific importance and relevant to the scientific work carried out within this manuscript. This would significantly shorten the manuscript, e.g. can section 2 be significantly shortened to possibly cover 1/2 page instead of the near 3 pages. The study area section can

also be shortened, e.g. is the tidal range is not important for the rest of the study. Similarly, is the last paragraph in section 3 not relevant for the presented work? Please revise the work bearing in mind what you are trying to convey and new scientific findings.

**A2:** The literature review on SAR backscattering from sea ice was shortened: highlights relevant for the manuscript were incorporated in the Introduction and in the Discussion. This, as well as the transfer of some content to the supplementary materials (as detailed in A11), has significantly shortened the manuscript.

**M2**:

Section "2. SAR backscattering over snow-covered sea ice", which presented a literature review on the seasonal evolution of C-band backscattering from first-year sea ice, was removed following your comments as well as those of Referee #1. Relevant references to the literature on this topic are now reserved for the discussion.

Information on the tidal range was removed from Section "3. Study area".

The paragraph detailing how the relevant local authorities were consulted and gave their approval for the project is still in the manuscript, because this information serves to demonstrate that the data was acquired in a respectful manner

**R3:** Please expand the methods section to explain to the reader what is done in the study and how the results are achieved. E.g. define and explain why the following is calculated; first freezing degree-day, freeze-up, frost flower maximum and winter onset?

**A3:** The methods section was expanded to include examples and definitions, and restructured to follow the manuscript objectives.

**M3:**

A paragraph explaining how the methods relate to the objectives of the manuscript was added as an introduction to the methods section:

> *"The objective of combining TerraSAR-X and time-lapse photography time-series for seasonal sea ice process observation is addressed by identifying indicators relating to freeze-up, winter, and breakup process elements. [...] Sections 4.1 and 4.2 describe the indicators and how they are observed or measured from each data source. Section 4.3 then explains how photographs are compared with coincident satellite images and used to identify their features, which serves to evaluate the potential of time-lapse photography to support TerraSAR-X image interpretation.*

In the Methods, we justify why each indicator is calculated by relating it to a process element, which is itself related to a given sea ice process:

> *"For example, elements of the freeze-up process include the date on which freeze-up begins, and the day on which it ends. These process elements can be observed through time-lapse photography indicators consisting of the first day where parts of the winter ice cover are observed on the water, and the first day where the winter ice cover is complete and stable."*

**R4:** According to the manuscript frost flowers could not be observed in the photographs, and as far as the reader can work out a peak in SAR backscatter values is therefore inferred to correspond to frost flowers. Though this is not again specifically stated in the methods section.

*Moreover, how do you know that frost flowers were present? Could the post-freeze-up peak be related to increased sea ice thickness?*

**A4:** We agree that the post-freeze-up peak cannot be reliably attributed to the presence of frost flowers. Indeed, frost flowers are too small to be resolved on the photographs. In general, following your comments and those of Referee #1, we now refrain from associating X-band backscattering time-series features to sea ice processes (e.g. frost flowers). Instead, the features are tracked as they are (e.g. post-freeze-up peak) throughout the Methods and Results. Potential explanations for the features based on physical processes (e.g. frost flowers, or increased sea ice thickness) are now reserved for the Discussion.

The difference between our observations of backscattering from nilas and those of the literature, which are several dBs lower, is suggested to be caused by the presence of a snow cover (snowfall was observed on the photographs). Frost flowers are cited as a possible additional source of scattering which cannot be resolved on the photographs (see excerpt below in M4). Regarding the specific case of the post-freeze-up peak, we were unable to confidently associate it with a physical process such as the presence of frost flowers or the growth of ice thickness, because of the relative lack of literature on X-band backscattering mechanisms within sea ice, particularly young sea ice in the winter. We do however offer potential explanations in the Discussion (see excerpt below in M4).

**M4**:

On the role frost flowers may have played in backscattering from nilas:

> *"In cold and dry snow conditions, the X-band isn't expected to penetrate significantly in the ice cover, with backscattering dominated by the presence of brine at the snow-ice interface (Nandan et al., 2016). Several factors may be intervening in backscattering from nilas. Frost flowers are known to increase the backscattering from newly formed sea ice in the C-band, an effect which may be more pronounced over thin ice; an increase of 5 dB was reported over ice 2 to 15 cm thick (Nghiem et al., 1997), and of 13 dB over 5 cm thick ice (Isleifson et al., 2014). Snow may also lead to an increase in backscattering through warming of the ice surface and brine absorption (Gill et al., 2015). In the case of our nilas observations, snow itself might be enough to explain the 3 dB difference; frost flowers may also have played a role, but cannot be observed on the photographs."* (in the Discussion)

On the mechanisms responsible for the post-freeze-up peak:

> *"The post-freeze-up peak and monotone backscattering onset are also observed in C-band time-series over sea ice (Yackel et al., 2007), but these seasonal features have been less studied than their spring counterparts (end of monotone backscattering and spring peak). Moreover, the same features in the X and C-band could well be related to different scattering mechanisms, and even to different physical processes. We limit ourselves to speculating, for the X-band data presented in this manuscript, that the increasing portion of the peak may be associated with the domination of surface scattering related to a brine-rich ice surface, potentially covered in frost flower, and that the decreasing portion may be associated with a transition to*

*an absorption regime, in which the signal suffers loss in the brine-wetted and increasingly colder snow. " (in the Discussion)*

**R5:** Why are not snow thickness information reported for any other time period than April 2017? In order to fully address the scientific topic indicated in the title of the paper information about the snow thickness is essential for the sea ice monitoring. It is unclear to the reader how are the estimates about the snow cover carried out? Photographs? Is there information about snow cover thickness and distribution? Judging by the title the manuscript should only contain information about snow-covered sea ice. Yet figures showing e.g., grease ice and pancake ice has no snow cover. Consider updating the title to reflect the sea ice included within the study. Please discus how do you expect the snow-cover to affect the results?

**A5:** The title of the manuscript and its objectives were reworded to better reflect the manuscript's focus on the combined use of TerraSAR-X and time-lapse photography for seasonal sea ice processes monitoring. Snow thickness measurements, performed by the authors as part of the greater Ice Monitoring project and included as context for the study site, have been updated to the entire available range of data (2015 to 2018).

The presence of a snow cover on the sea ice, which is the case for almost all the TerraSAR-X images in the time-series, is expected to play a significant role in the total backscattering and its seasonal evolution. Following freeze-up, the snow absorbs brine expelled from the new ice (Barber and Nghiem, 1999). The presence of brine in the snow is expected to restrain the interaction of the signal to the snow volume and the snow-ice surface because of absorption in the lossy brine-wetted snow (Nandan et al., 2016). Penetration in the ice cover itself is therefore small (modelled penetration depth is for the X-band VV is 2 cm in brine-wetted snow, as reported by Nandan et al., 2016). Relevant scattering mechanisms should be limited to surface and volume scattering within the snow layers and surface scattering at the snow-ice interface.

**M5**:

Reworded Title:

*"Combining TerraSAR-X and time-lapse photography for seasonal sea ice monitoring: the case of Deception Bay, Nunavik" (Title)*

Reworded manuscript objectives:

*"This article explores the use of combined TerraSAR-X and time-lapse photography time-series to observe seasonal sea ice processes, and the potential of the time-lapse photography to support TerraSAR-X interpretation. The case study is performed over three years in Nunavik's Deception Bay. A complementary objective is to describe the processes through an interannual comparison. (in the Introduction)*

Updated snow thickness data and provided details and a reference regarding the measurement method:

*"Point thickness measurements performed in Deception Bay for the Ice Monitoring project in January-February and April-May 2016, 2017, and 2018 (Gauthier et al., 2018) ranged from 0 to 55 cm for the snow cover, and 52 to 165 cm for the ice cover. " (in the Study area)*

**R6:** Why is beta nought used instead of the more commonly used sigma nought? Since beta nought is used please use the symbol beta in all the figures 6, 10, 11, 12 and 16 instead of the symbol sigma.

**A6:** In the Methods section, it was incorrectly indicated that the TerraSAR-X data had been processed in beta-naught. The data is actually in the conventional sigma-naught, which is why the sigma symbol is used throughout the manuscript.

**M6**: Corrected:

> *"This workflow starts with a conversion from the digital number to radar brightness (sigma-naught) [...]." (in the Methods)*

**R7:** The incidence angle difference within the dataset is significant (8o), how does this affect the presented results?

**A7:** The seasonal features (peaks and inflexion points) were observed for all acquisition parameters, so the different incidence angles between orbits doesn't affect the results presented in the manuscript. The relative amplitude of the features did however sometimes depended on incidence angle. To discuss the effect of incidence angle, we compare the two ascending-evening orbits. Orbit 13 (empty squares) at an incidence angle of 38° and orbit 89 (black diamonds) at an incidence angle of 46° are first shown for 2017-2018 (see Fig. R2-1).

[Figure]

**Figure R2-1**: TerraSAR-X median VV backscattering plotted versus time for 2017-2018. Two orbits are shown: orbits 13 (38°, empty square) and 89 (46°, black diamond).

A small incidence angle effect can be seen on the median VV backscattering during the post-freeze-up and spring peaks. The backscattering at 38° is 1 to 3 dB higher than the backscattering at 46°. No incidence angle effect is seen in the monotone winter period. The same is observed in 2016-2017 (see Fig. R2-2). A backscattering signal which decreases with incidence angle is expected for situations dominated by surface scattering on a relatively rough surface (Ulaby et al., 1986). Surface scattering at the interfaces between dry snow, brine-wetted snow and ice is indeed expected to dominate backscattering from cold snow-covered sea ice, with a transition to mixed scattering for thicker snow covers (Gill et al., 2015). The monotone winter period where no incidence angle effect is observed might be associated with mixed scattering.

[Figure]

**Figure R2-2**: TerraSAR-X median VV backscattering plotted versus time for 2016-2017. Two orbits are shown: orbits 13 (38°, empty square) and 89 (46°, black diamond).

The effect of incidence angle is completely different in 2015-2016 however. There is an almost constant difference of 2 dB between the two orbits, with the backscattering at 38° always **lower** than the 46° backscattering (see Fig. R2-3). The freeze-up process was also different in 2015: the thermal freeze-up is expected to have produced smooth thermal ice, compared to rougher ice from nilas patches and other ice types in 2016 and 2017. We suggest that, for 2015-2016, the surface scattering at the snow-ice interface was consistently low because of a smoother ice cover, and that the total backscattering was dominated instead by volume scattering. Volume scattering may increase with incidence angle (Ulaby et al., 1986), as observed in the 2015-2016 data throughout the winter and spring.

[Figure]

**Figure R2-3**: TerraSAR-X median VV backscattering plotted versus time for 2015-2016. Two orbits are shown: orbits 13 (38°, empty square) and 89 (46°, black diamond).

**M7**: A discussion on the effect of incidence angle has been added to the discussion:

*"Before moving on to the spring processes, we first discuss the influence of an 8° difference between ascending orbits 13 and 89. For 2016-2017 and 2017-2018, a small incidence angle effect was seen during the post-freeze-up and spring peaks, where backscattering was 1 to 3 dB smaller at the higher incidence angle, and no effect was seen during the monotone winter period (see Fig. 9 and 12). A backscattering signal which decreases with incidence angle is expected for situations dominated by surface scattering on a relatively rough surface (Ulaby et al., 1986). In the C-band, surface scattering at the interfaces between dry snow, brine-wetted snow and ice is indeed expected to dominate for cold snow-covered sea ice, with a transition*

*to mixed scattering for thicker snow covers (Gill et al., 2015). We speculate that surface scattering on the ice formed from nilas patches explains the dependence on incidence angle observed in our X-band data. 2015-2016 however presents a very different case. Backscattering at the higher incidence angle is consistently 2 dB stronger than at the lower incidence angle, throughout winter and during the spring peak (see Fig. 12). We've shown the freeze-up process to have been different that year compared to 2016 and 2017, and already suggested that the ice cover was much smoother for the 2015-2016 season. We speculate that surface scattering was consistently low that year, and that volume scattering, which Ulaby et al. (1986) have shown can increase with incidence angle, dominated instead." (in the Discussion)*

**R8:** The fjord is given as 20 km wide (a scale bar would be nice to see in Figure 2), how much of the overall area is covered by the 15km wide TerraSAR-X images?
**A8:** A scale bar was added to Fig. 2, and the overlap between the study area and the TerraSAR-X image subset (a 9 km long section of the bay) is now indicated in the description of the study site and the TerraSAR-X data.
**M8**:
Added a scale bar to the map of Deception Bay.
Added detail on study area coverage by the TerraSAR-X images:
> *"Figure 2 shows the extent of the subimages, which overlaps with a 9 km long section of the bay." (in Data description)*

**R9:** It is in the discussion stated that the sea ice observed in the ship wake was broken. How is this verified? How are wind effects accounted for in these observations? Would it be possible to include observations from these ships?
**A9:** Following their transit in the bay, the *MV Nunavik* and *MV Arctic* leave a track of open water and floating broken ice pieces. Because the ice cover is landfast to both shores, the track remains "open" until it has refrozen (in the winter); it is never closed by wind or currents. Ice lateral movement under the effect of the wind could be resolved on time-lapse photography, and was not observed. The  refreezing of the track will depend on how much broken ice is in the open water and on the air temperature. The broken ice left in the ship's wake can be observed on time-lapse photography for the areas close to the cameras. We also observed the tracks left by the ships during winter fieldwork in Deception Bay in 2016, 2017, and 2018. During the spring, when ice-breaking transport resumes around June 1st, the tracks left by the ship are unlikely to refreeze due to warm temperatures. We speculate that the ship tracks are then left open, potentially with floating ice pieces depending on the ice and meteorological conditions. We don't have access to observations from the ships.
**M9**: Following the rewrite, the particular sentence which referred to broken ice left in the ship's wake is no longer in the discussion.
The effect of the wind (or absence thereof) has been added to the discussion on breakup processes:
> *"We speculate that the ice cover was in a more advanced state of degradation when breakup started in 2016 than in 2017. This is supported by time-lapse photography*

*which show that the ice cover was partly mobile (under the effect of wind or current) during breakup 2016, but mostly landfast during breakup 2017 (Movies S4-S5). In 2018's comparatively late spring, both the MV Nunavik and MV Arctic entered the bay during pond onset (on June 17th). Open water was observed along their tracks in the following days, and new cracks perpendicular to the shore appeared when the ships left the bay eight days later." (in the Discussion)*

**R10:** It is in the manuscript stated that the X-band backscatter change is expected to be similar to the one in C-band. A suggested to corroborate this is to include Sentinel-1 images overlapping the fjord to investigate if these C-band images show the same evolution as the TerraSAR-X images. The use of Sentinel-1 may also reduce the revisit time.

**A10:** Following comments from Referee #1, we removed the assumption that the seasonal evolution of the X-band backscattering should be similar to that of the C-band. The question of similarity between the two bands is now reserved for the discussion.

During the Ice Monitoring project, RADARSAT-2 C-band data was acquired during winters (December to April) 2015-2016, 2016-2017, and 2017-2018, in partnership with the Canadian Ice Service. The median VV RADARSAT-2 backscattering at 36° over the same AOIs as presented in the paper is shown in Fig. R2-4, alongside TerraSAR-X data from orbit 13 at 38°. The RADARSAT-2 data was processed in the same way as the TerraSAR-X data, using the Multi-SAR-System at DLR.

[Figure]

**Figure R2-4**: TerraSAR-X (38°) and RADARSAT-2 (36°) median VV backscattering plotted versus time. RADARSAT-2: in full diamonds and TerraSAR-X in empty squares.

The available RADARSAT-2 data closely matches the TerraSAR-X time-series for 2016-2017 and 2017-2018 monotone winter periods, as well as for the 2017 acquisition in the

post-freeze-up peak. No data is available during one of the spring peaks. In 2015-2016, the monotone winter period backscattering is consistently lower in the C-band than in the X-band, by approximately 4 dB, which is probably due to differences in penetration depth. As described in A7, ice in 2015-2016 was presumably smoother than the other two years. The low signal both in X and C-band (Fig. R2-4) suggests that the ice appeared smooth at both radar frequencies. The C-band's increased penetration depth (due to a longer wavelength) might have led to increased absorption of the signal by brine inclusions in the sea ice, leading to a lower total backscattering than at the X-band.

**M10**: Added to the conclusion:

> *"Future work in the Ice Monitoring project will build on this characterization of seasonal processes and focus on spatial variations within the bay and comparison with similar fjords, namely Salluit and Kangiqsujuaq. It will also involve comparison of the TerraSAR-X time-series data with RADARSAT-2 time-series acquired over the same period and area." (in the Conclusion)*

**R11:** Rather than stating that the standard meteorological station at the airport is not used, a correlation measure between the used temperature and the airport temperature would have been beneficial. Such a comparison would also have verified the supposedly heated camera case claim, rather than a statement that the authors think that it is so. At what altitude is the temperature sensor located?

**A11**: We performed a correlation analysis between the camera temperature measured in Deception Bay and the airport temperature measured in Salluit, 50 km away. Everything related to the temperature measurements has been moved to a supplementary document focused on air temperature. It includes a description of the two different temperature data sources (including altitude), the data acquisition method, and a comparison of the two datasets (including the Pearson coefficient). The airport data was shown to be strongly correlated to the Deception Bay camera measurement. We therefore chose to use the airport data, despite the 50 km distance, to document how warm or cold each month was. We removed the bias modeling and correction for the camera data. Results on monthly cumulative freezing and thawing degree-days are given in the supplementary document for the airport dataset, and cited at several points in the Discussion.

**M11**:

Added detail to the description of the temperature data:

> *"The closest weather station to Deception Bay is located in neighboring Salluit, at the airport. Measurements at the station are taken hourly during the day, and their daily mean is available online from Environment Canada. Salluit is a Nunavik coastal community located 50 km west of Deception Bay; the airport is located 2.8 km inland at an altitude of 226 m.*
>
> *Two Reconyx PC800 Hyperfire Professional Semi Covert cameras were installed in Deception Bay as part of the CAIMAN research project. These cameras were installed*

*near the study area, in front of Moosehead Island at an altitude of 22 m (series A) and on Black Point at an altitude of 33 m (series B)." (in the Supplementary)*

Added a correlation analysis:

*"Figure S3-2 shows the camera and airport datasets from 2015 to 2018. The camera and airport measurements showed a Pearson coefficient of 0.98 and 0.99 for the three years of the study, which proves a strong correlation despite their different locations. The camera dataset however differs from the airport's dataset with a root mean-squared-error (RMSE) of 3.9 to 4.3°C. To investigate this discrepancy, Fig. S3-2 also shows that the daily difference between the two datasets is roughly flat from September to January, and then peaks in April-May.*

[Figure]

***Figure S3-2**: Left: Daily mean air temperature measured by the camera in Deception Bay (black) and at the Salluit airport (red). Right: Daily difference between the two datasets." (in the Supplementary)*

**R12**: As stated in the manuscript is FDD used, yet the unit used was oC, please explain? Also freezing at +3.5 oC to +4oC seems a bit high. Why is 0oC used when sea ice is investigated as it will likely freeze at -1.8oC. Are there any sea water temperature measurements?

**A12**: Freezing degree-days are a sum of temperatures (measured in °C). Since summation preserves units, FDDs were presented with a °C.

Freezing degree-days are a sum of freezing temperatures normalized to a positive number; freezing degree-days of 3.5 and 4.5 °C on average therefore mean that the daily temperature was roughly -4°C on average between the first freezing day of the year and the day of freeze-up, which is coherent with water freezing at -1.8°C.

While it is true that sea water freezes at slightly lower temperatures than fresh water, we chose to use the conventional definition of freezing and thawing degree-days (relative to the 0°C

mark), used by the Ouranos consortium in their climate projections for the region for instance (Mailhot and Chaumont, 2017).

No surface sea water temperature measurements were made as part of the fieldwork for this project. Temperature measurements reported in the literature were taken in September 2006 (GENIVAR, 2007) and August 2012 (GENIVAR, 2012).

**M12**: Added the reference to the Ouranos consortium climate projections to the description of freezing and thawing degree-days.

**R13:** It is mentioned in the discussion that the Deception River has warm water, please provide a temperature time series for this river or at least give some specific temperatures.

**A13:** No water temperature measurements were performed for Deception River, and we found no such data in the literature for the breakup period.

**M13**: The statement on Deception River having warm or warmer water was removed from the discussion.

**R14:** How do the values in e.g. Fig. 10 relate to e.g. work by Onstott 1992?

**A14:** We added a comparison of our results with those presented by Onstott (1992) and others.

**M14**: Added:

> *"Despite a difference of almost 20° in the incidence angle, our observation of -12 ± 1 dB over unidentified ice one to nine days after freeze-up (Fig. 8) is close to [...] reports by Onstott (1992) of -14.4 dB over thin first year ice (30 to 70 cm thick) in the X-band HH at 23°."* (in the Discussion)

**R15:** Additional comments According to the temperature records, as presented on row 254, temperatures are recorded between 11 September 2015 – 16 September 2016 and then from 18 September 2018 (?) to 31 August 2018. Please clarify.

**A15:** This is a mistake; the correct date ranges are 11 September 2015 – 16 September 2016 and 18 September 2016 – 31 August 2018.

**M15**: Corrected in the manuscript.

**R16:** Ibid. is not a way to reference that I've seen in this journal before. Whilst this might still be ok, many of the references where ibid. has been used are not correct and the statements that are supposed to be covered in those references are not included here. E.g. on row 129 the "white ice" term is attributed to Johansson et. al., 2017 and on row 400 correctly to WMO. Moreover, when referring to the WMO terminology please include the full reference, (WMO, 2014).

**A16:** We removed all uses of "ibid" and checked all citations.

**M16**: All uses of "ibid" were changed to in-text citations.

**References**

Barber, D. G. and Nghiem, S. V.: The role of snow on the thermal dependence of microwave backscatter over sea ice, J. Geophys. Res.-Oceans, 104(C11), 25789–25803, https://doi.org/10.1029/1999JC900181, 1999.

Gauthier, Y., Dufour-Beauséjour, S., Poulin, J., and Bernier, M. 2018. Ice Monitoring in Deception Bay : Progress report 2016-2018. Québec: INRS, Centre Eau Terre Environnement. http://espace.inrs.ca/7538/.

GENIVAR: Projet Nickélifère Raglan Sud – Rapport principal – Étude d'impact sur l'environnement et le milieu social. Rapport de GENIVAR Société en commandite pour Canadian Royalties inc. Montreal, Que.,2007.

GENIVAR: Environmental and Social Impact Assessment of the Deception Bay Wharf and Sediment Management, Report from GENIVAR for Canadian Royalties Inc., Montreal, Que., 2012.

Gill, J. P. S., Yackel, J. J., Geldsetzer, T. and Fuller, M. C.: Sensitivity of C-band synthetic aperture radar polarimetric parameters to snow thickness over landfast smooth first-year sea ice, Remote Sens. Environ., 166, 34–49, https://doi.org/10.1016/j.rse.2015.06.005, 2015.

Isleifson, D., Galley, R. J., Barber, D. G., Landy, J. C., Komarov, A. S. and Shafai, L.: A Study on the C-Band Polarimetric Scattering and Physical Characteristics of Frost Flowers on Experimental Sea Ice, IEEE Transactions on Geoscience and Remote Sensing, 52(3), 1787–1798, https://doi.org/10.1109/TGRS.2013.2255060, 2014.

Mailhot, A. and Chaumont, D.: Élaboration du portrait bioclimatique futur du Nunavik - Tome II. Rapport présenté au Ministère de la forêt, de la faune et des parcs., Ouranos, Montreal, Que., 2017.

Nandan, V., Geldsetzer, T., Islam, T., Yackel, John. J., Gill, J. P. S., Fuller, Mark. C., Gunn, G. and Duguay, C.: Ku-, X- and C-band measured and modeled microwave backscatter from a highly saline snow cover on first-year sea ice, Remote Sens. Environ., 187, 62–75, https://doi.org/10.1016/j.rse.2016.10.004, 2016.

Nghiem, S. V., S. Martin, D. K. Perovich, R. Kwok, R. Drucker, and A. J. Gow. 1997. "A Laboratory Study of the Effect of Frost Flowers on C Band Radar Backscatter from Sea Ice." Journal of Geophysical Research: Oceans102 (C2): 3357–70. https://doi.org/10.1029/96JC03208.

Onstott, R. G.: SAR and Scatterometer Signatures of Sea Ice, in: Microwave Remote Sensing of Sea Ice, American Geophysical Union, Washington, D.C., United States, 73-104, 1992.

Ulaby, F. T., R. K. Moore, and A. K. Fung. 1986. Microwave Remote Sensing: Active and Passive. Vol. Volume 3-From theory to applications. United States: Addison-Wesley Publishing Company. https://ntrs.nasa.gov/search.jsp?R=19860041708.

Yackel, J. J., Barber, D. G., Papakyriakou, T. N. and Breneman, C.: First-year sea ice spring melt transitions in the Canadian Arctic Archipelago from time-series synthetic aperture radar data, 1992–2002, Hydrol. Process., 21(2), 253–265, https://doi.org/10.1002/hyp.6240, 2007.

---

## Author Response (AR2)

**Answers to referees**

Thank you very much for your comments and suggestions on this version and previous version of our manuscript. They were very helpfull to clarify our research objectives, to discuss our results and improve the overall manuscript.

Sincerely yours,

The authors

**Anonymous Referee #2**

We incorporated all your suggestions. Below, we reproduced your comments (R), provided our answers (A), and detailed changes to the manuscript (M).

**R1:** L28-29 ... "The evolution of seasonal sea ice conditions in Deception
Bay should continue ... " ... is an odd/awkward sentence. What are you really trying to say here? Pls revise.
**A1:** The sentence was rewritten.
**M1**:
L28-30 *"Seasonal sea ice conditions in Deception Bay will continue to evolve: climate projections for the region include shorter snow cover periods and warmer annual average temperature in 2040-2064 (Mailhot and Chaumont, 2017)."*

**R2:** L43 ... What is meant by .. "Necessary to rely [both] on spatial coverage of the bay ..". It is not clear.
**A2:** The sentence was rewritten.
**M2**:
L48-52 *"The sequence of events may vary from one area to another, influenced by geomorphological features like shallows or deep water pockets, islands, and rivers. To capture the spatio-temporal nature of these processes, their observation should therefore integrate both spatial coverage and frequent observations. The combined use of radar remote sensing and time-lapse photography meets these requirements."*

**R3:** L49. Change from SAR X-band to X-band SAR.
**A3:** The sentence was rewritten as suggested
**M3:**
L56 *"X-band SAR has been shown to be a useful complement to the conventional C-band when it comes to first year sea ice: it was used to identify types of new ice (Johansson et al., 2017), particularly thin ice like nilas and grey ice (Matsuoka et al., 2001)."*

**R4:** L56. The authors should also cite one the Nandan et al., 2017 papers in either Remote Sensing or Remote Sensing of Environment dealing with the use of X-band in comparison to C-band.

**A4:** A reference to Nandan et al., 2017 in Remote Sensing was added in the Introduction and in the Discussion.

**M4:**

L63 *"They include observations over new ice and nilas (Johansson et al., 2017; 2018) and white ice (Fors et al., 2016), as well as over first-year sea ice during the spring (Nandan et al., 2016; Nandan et al., 2017; Paul et al., 2015)."*

L368 *"Indeed, despite their spectral proximity, the C-band and X-band have been shown to behave differently when it comes to interaction with brine-wetted snow for instance (Nandan et al., 2016; Nandan et al., 2017)."*

**R5:** L80-81 and L83-89. These are not objective-type statements. They need to be removed from this section. If you want to put L83-89 into the 'context' section of the Introduction then that is the place for it.

**A5:** The Context and Objectives sections of the Introduction were reorganised as suggested.

**M5:**

L35-43, in Context: *"Raglan Mine initiated this project [...]. Finally, the Nunavik Marine Region Impact Review Board gave permission for the deployment of underwater sonars in Deception Bay (sonar data not presented in this article). Local sea ice monitoring is relevant in light of local community members' reliance on the fjord's rich ecosystem for subsistence, as well as for shipping-related operations by the mines. More generally, this case study stands out due to the length of the time-series reported, and may be useful to those wishing to monitor seasonal processes in remote areas or interested in sea ice processes."*

L84-87, in Objectives: *"This article explores the use of combined TerraSAR-X and time-lapse photography time-series to monitor seasonal sea ice processes and the potential of time-lapse photography to support TerraSAR-X interpretation. We performed this case study over three years in Nunavik's Deception Bay. A complementary objective is to describe the processes through an interannual comparison."*

**R6:** L139. INRS is first mentioned here and needs to be spelled out.

**A6:** The sentence was rewritten as suggested

**M6:**

L138-139 *"Photographs are automatically transferred through Raglan's network to a database hosted by Institut national de la recherche scientifique (INRS)."*

**R7:** A final general comment: Since your TSAR-X data spans an incidence angle range of 8 degrees between the various years, this, as you have acknowledged, can lead to varying amounts of surface scattering relative to volume scattering. I suggest you comment briefly in your TSAR-X processing section 4.1 on the effect of incidence angle towards the potential variability of the time series observed in your various monotone periods. A key paper to reference here, albeit at C-band, is Mahmud et al., 2018 IEEE TGARS.

**A7:** Details on the variation in incidence angle range within a single image of a given orbit were added to the TerraSAR-X processing section 4.1. In the Discussion, our results on the difference in backscattering between images acquired at different incidence angles were compared with those of Mahmud et al., 2018.

**M7:**

L161-162, in 4.1 *"The TerraSAR-X noise floor for the three orbits ranges between -23 and -24.5 dB, the difference between maximum and minimum incidence angle within an image ranges from 1.4° to 1.0°, and the radiometric accuracy is 0.6 dB (Eineder et al. 2008)."*

L435-439, in the Discussion *"Mahmud et al. (2018) recently modeled the linear decrease with incidence angle of L- and C-band HH backscattering (in dB) from first-year ice; their results show a dependence of -0.22 dB/1°. This would translate as a 1.8 dB for an 8° difference in the C-band and HH polarization, which is similar to what we observe in the X-band and VV polarization for 2016-2017 and 2017-2018."*

**List of changes**

During minor revisions, the following content changes were made to manuscript tc-2019-199:

1. Introduction: Some content was moved from the Objectives section to the Context section.

3. Data description: Details on the incidence angle range within a given TerraSAR-X image were added.

6. Discussion: A comment was added to the discussion relating to the impact of an 8 degree incidence angle difference on backscattering.

Other minor changes were made such as rewording, and are listed in the detailed point-by-point response to the reviewers.

Sophie Dufour-Beauséjour, for the authors
sophie.dufour-beausejour@ete.inrs.ca

[revised manuscript text omitted]
 also to the anonymous reviewers whose comments greatly improved this article, and to the editorial team. 
[revised manuscript text omitted]